# Inferential eye movement control while following dynamic gaze

**Nicole Xiao Han[1]\*, Miguel Patricio Eckstein[2]**

[1]Department of Psychological and Brain Sciences, Institute for Collaborative Biotechnologies, University of California, Santa Barbara, Santa Barbara, United States; [2]Department of Psychological and Brain Sciences, Department of Electrical and Computer Engineering, Department of Computer Science, Institute for Collaborative Biotechnologies, University of California, Santa Barbara, Santa Barbara, United States

**Abstract** Attending to other people's gaze is evolutionary important to make inferences about intentions and actions. Gaze influences covert attention and triggers eye movements. However, we know little about how the brain controls the fine-grain dynamics of eye movements during gaze following. Observers followed people's gaze shifts in videos during search and we related the observer eye movement dynamics to the time course of gazer head movements extracted by a deep neural network. We show that the observers' brains use information in the visual periphery to execute predictive saccades that anticipate the information in the gazer's head direction by 190–350ms. The brain simultaneously monitors moment-to-moment changes in the gazer's head velocity to dynamically alter eye movements and re-fixate the gazer (reverse saccades) when the head accelerates before the initiation of the first forward gaze-following saccade. Using saccade-contingent manipulations of the videos, we experimentally show that the reverse saccades are planned concurrently with the first forward gaze-following saccade and have a functional role in reducing subsequent errors fixating on the gaze goal. Together, our findings characterize the inferential and functional nature of social attention's fine-grain eye movement dynamics.

\*For correspondence:
xhan01@ucsb.edu

**Competing interest:** The authors declare that no competing interests exist.

## Editor's evaluation

This important work substantially advances our understanding of how human eye movements are shaped by social cues. Using clever experimental manipulations and innovative artificial intelligence analysis tools, the paper identifies distinctive patterns of saccadic eye movements tracking another person's gaze during dynamic video-scene viewing. This work will be of broad interest to psychologists, biologists, and neuroscientists interested in human social behavior.

## Introduction

Eye movements are involved in almost every daily human activity, from searching for your apartment key, identifying a friend, reading, and preparing a sandwich. People make about three ballistic eye movements (saccades) per second orienting the central part of the vision (the fovea) to regions of interest in the world and acquiring high-acuity visual information (*Bahill et al., 1975*). A foveated visual system allocates more retinal cells (*Curcio et al., 1987*), thalamic, cortical (*Azzopardi and Cowey, 1993*), and superior colliculus (*Chen et al., 2019*) neurons to the central part of the visual field and allows for computational and metabolic savings. But it requires that eye movements are programmed intelligently to overcome the deficits of peripheral processing. People execute eye movements effortlessly, rapidly, and automatically giving the impression that these might seem fairly random. However,

there are sophisticated computations in the brain that control eye movements involved in fine spatial judgments (*Intoy and Rucci, 2020*; *Rucci et al., 2007*), search (*Araujo et al., 2001*; *Eckstein et al., 2015*; *Hoppe and Rothkopf, 2019*; *Najemnik and Geisler, 2005*), object identification (*Renninger et al., 2007*), face recognition (*Or et al., 2015*; *Peterson and Eckstein, 2012*), reading (*Legge et al., 1997*; *Legge et al., 2002*), and motor actions (*Ballard et al., 1995*; *Hayhoe and Ballard, 2005*).

Following the gaze of others (gaze-following) with eye movements is critical to infer others' intentions, current interests, and future actions (*Capozzi and Ristic, 2018*; *Dalmaso et al., 2020*; *Emery, 2000*; *Kleinke, 1986*; *McKay et al., 2021*). Gaze-following behavior can be observed as early as 8–10 months in infants and is widely found in nonhumans such as macaques, dogs, and goats (*Brooks and Meltzoff, 2005*; *Kaminski et al., 2005*; *Senju and Csibra, 2008*; *Shepherd, 2010*; *Wallis et al., 2015*). The ability to perceive others' gaze direction accurately and plan eye movements is essential for infants to engage in social interactions to learn objects and languages (*Carpenter et al., 1998*; *Morales et al., 1998*; *Morales et al., 2000*; *Woodward, 2003*). People are extremely sensitive to others' direction of gaze (*Kleinke, 1986*; *Langton and Bruce, 1999*). When people observe someone's gaze, they orient covert attention and eye movements toward the gazed location, which improves the detection of a target appearing in the gaze direction (*Driver et al., 1999*; *Egeth and Yantis, 1997*; *Friesen et al., 2004*; *Han et al., 2021b*; *Jonides, 1981*; *Kingstone et al., 2003*; *Mulckhuyse and Theeuwes, 2010*). Impairments in gaze cueing have also been proposed as an important correlate of autism spectrum disorder (*Baron-Cohen, 2001*; *Leekam et al., 1998*; *Nation and Penny, 2008*) and important in child development (*Brooks and Meltzoff, 2005*).

The majority of studies investigating gaze-following (but see *Gregory, 2022*; *Han et al., 2021b*; *Lachat et al., 2012*; *Macdonald and Tatler, 2013*; *Sun et al., 2017*; *Wang et al., 2014*) use static images of the eyes or the face in isolation, which are far from the more ecological real-world behaviors of individuals moving their heads and eyes when orienting attention. That gaze cueing triggers eye movements is well known, but the dynamics of eye movements when observing gaze behaviors with naturalistic dynamic stimuli are not known. Studies have investigated how the brain integrates temporal information to program saccades and how it integrates foveal and peripheral information (*Stewart et al., 2020*; *Wolf et al., 2022*; *Wolf and Schütz, 2015*) but have relied on artificial or simplified stimuli.

Little is known about what features across the visual field influence eye movements during gaze-following, their temporal dynamics, and their functionality. How does the brain rely on the features of the gazer's head and peripheral visual information about likely gaze goals to program eye movements? Do observers wait for the gazer's head movement to end before initiating the first gaze-following saccades? Do visual properties of the gazer's head influence the programming of eye movements? Answering these questions has been difficult because they require a well-controlled real-world data set, moment-to-moment characterization of the gazer's features, and experimental manipulations that alter peripheral information while maintaining the gazer's information unaltered.

Here, we created a collection of in-house videos of dynamic gaze behaviors in real-world settings by instructing actors to direct their gaze to specific people on the filming set (*Figure 1a*). We then asked experiment participants that watched the videos to follow the gaze shifts in the videos and decide whether a specific target person was present or absent (*Figure 1a*). The viewing angles subtended by the people in the videos corresponded to distances for which the eyes provided little information. Thus, our work focuses on the gazer's head direction.

Our first goal is to assess the impact of peripheral gaze-goal information on the saccade error and timing. Second, we aimed to elucidate how the brain temporally processes visual information to influence saccade programming during gaze-following. To extract features of the videos that we hypothesized would influence saccade planning we used a state-of-the-art artificial intelligence (AI) model (*Chong et al., 2020*) to make moment-to-moment estimates of the gazer's head direction in the videos. We assessed how observers' saccade direction, timing, and errors related to the extracted features to gain insight into the brain computations during saccade planning.

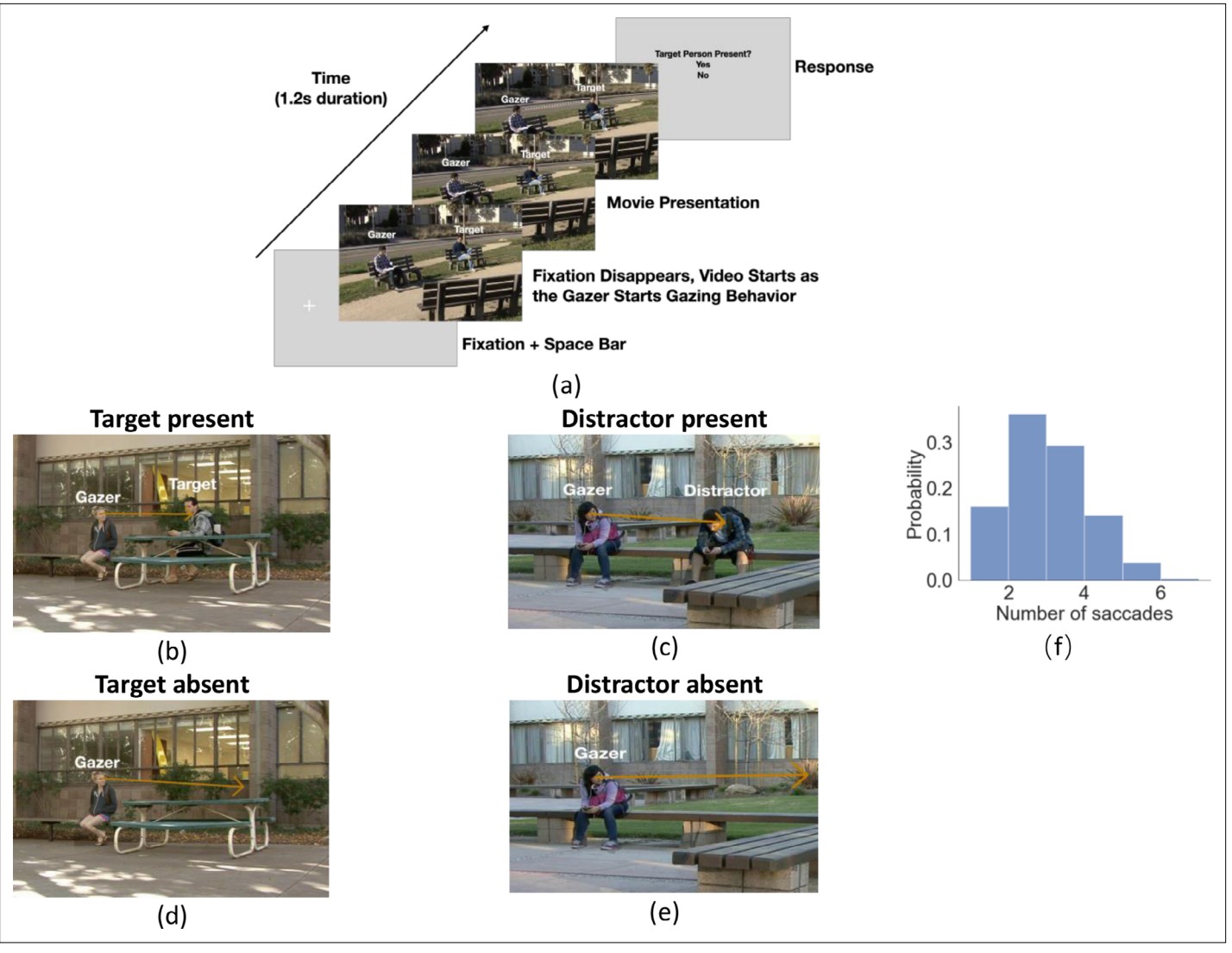

**Figure 1.** Experiment workflow and example trials. (**a**) Timeline for each trial. The participants fixated on the fixation cross and pressed the space bar to initialize the trial. The cross was located at the gazer's head, and the trial would not start if the eye fixation moved away from the cross by 1.5°. The cross disappeared as the video started with the gazer initiating their head movement to look at the designated gazed person (25% target present and 25% distractor present and 50% gaze goal absent). Participants were instructed to follow the gaze of the person in the video, which lasted 1.2 s. After the end of the video, a response image was presented and the observers selected 'Yes' or 'No' to respond to whether the target person was present or not in the video. The target person was the same throughout the trials but appeared with different clothing across trials. (**b-e**) Example video frames of the gazer looking at the gaze goal (distractor or target) either with the person present (**b, c**) or absent (digitally deleted, **d, e**). The orange arrow vector is the gaze estimation from a deep neural network model, with details presented in the following section. Note that all the text annotations and arrows are just for illustration purposes and were not presented during experiments. (**f**) Histogram of the number of saccades participants executed per trial.

The online version of this article includes the following figure supplement(s) for figure 1:

**Figure supplement 1.** Histogram of gazed person eccentricities relative to the gazer's position.

## Results

### Integration of peripheral information to guide gaze-following saccades

An observer looking at a gazer can use different sources of visual information to estimate where a gazer is looking at. There are instances in which a gazer looks at a large object (e.g. a person, a car, etc) and the observer foveating on the gazer can use both the gazer's head or body movements and the likely gazer goals in the observer's visual periphery to make inferences about where the gazer is attending. In other circumstances, the gazer looks at a small object (e.g. a small key) on the floor or placed on a long table and the observer foveating at the gazer might have to rely only on

the gazer's head direction information because the small object is not visible in the observer's visual periphery. In our study, we used digital editing tools to erase potential gaze goals while maintaining the gazer's movements unaltered and preserving the video's background (*Figure 1b–e*). This manipulation allowed us to isolate the influence of the gazer's head movements and that of peripheral gaze-goal information on the observers' eye movements. Eliminating the gaze goal can be interpreted as mimicking a scenario in which the gaze goal is very small and not visible in the observer's visual periphery. If observers heavily relied on the gazer's head movements and did not use peripheral information, removing the gaze goals will have little impact on the saccade errors. Similarly, if motor programming error (*van Beers, 2007*; *Han et al., 2021a*) represents the bottleneck in the precision of the saccade endpoint, then the digital deletion of the peripheral gaze goal will also have little impact. On the other hand, if observers integrate the gazer's head information with the peripheral visual information to guide their gaze-following saccades then the elimination of the gaze goal will increase the observer's saccade error. We separately analyzed the saccade endpoint angular, amplitude and Euclidean error (distance from the saccade endpoint and the gaze goal location). A priori, we might expect the peripheral visual information about the gaze goal to be more critical to reduce the saccade endpoint Euclidean error. The gazer's head direction might be sufficient for the brain to program saccades with an accurate angular direction. Thus, we might expect saccade angular error to be less influenced by the peripheral presence/absence of the gaze goal.

Twenty-five observers viewed 80 in-house videos (1.2 s long, different settings) of an actor (gazer) actively shifting his/her head and gaze to look at another person (gaze goal) in the video. Participants' initial fixation was on the gazer's head. They were instructed to look where the gazer looks and report whether a specific target person was the gaze goal (*Figure 1a* for a timeline of stimuli). In 25% of the videos, the target person was present and always the gaze goal (*Figure 1b*). The target person was the same across all videos but might appear in different clothing. In another 25%, a distractor person (*Figure 1c*) was the gaze goal and the target person was absent. In the remaining 50%, no person was at the gazed location (d-e). The target/distractor-absent (no person) videos were created by digitally removing the person at the gaze-goal location while preserving the immediate background. The gazers' visual information in the videos was identical in the target/distractor-present vs. absent videos (*Figure 1b* vs. 1d and 1 c vs. 1e). Throughout the trial, we measured eye position and detected the onset of saccades registered to the video timing. Observers typically executed 3–5 saccades. *Figure 1f* shows a histogram of the number of executed saccades per trial.

To investigate the effect of peripheral information on eye movement planning, we tested the influence of the presence of a person at the gaze goal on the first saccade error and timing. *Figure 2a–d* show heat maps of first saccade endpoint distributions across all observers (for one particular video) and illustrate how the peripheral presence of a person at the gaze goal reduces the error of the first fixation. To quantify the error we calculated the saccade angular error (degrees), saccade amplitude error (degrees of visual angle °), and saccade Euclidean error (degrees of visual angle °) which was a combination of angular and amplitude error. We found a significant main effect of the presence of a person at the gaze goal location for saccade angular error (2 (present or absent) x 2 (target or distractor) ANOVA, $F_{(1,24)} = 100$, p = 4.85e-10, $\eta^2 = 0.987$, *Figure 2e*). The angular error was higher when the person at the gaze goal was absent target: 57.4 degrees for absent vs. 32.8 degrees for present; distractor: 54.9 degrees for absent vs. 29.3 degrees for present. We also found that the presence of a person at the gaze goal in the periphery reduced the amplitude error (*Figure 2e*, 2 (present or absent) x 2 (target or distractor) ANOVA, $F_{(1,24)}=207$, p = 2.7e-13, $\eta^2 = 0.99$). The amplitude error was higher when a person was absent vs. present for both target (*Figure 2e*, absent 4.23° vs. present 1.87°, p = 3.27e-119 with FDR) and distractor (absent 4.43° vs. present 1.98°, p = 1.1e-144 with FDR). It also reduced the Euclidean error that combines the angular and amplitude error (2 (present or absent) x 2 (target or distractor) ANOVA, $F_{(1,24)}=259$, p = 2.3e-14, $\eta^2 = 0.99$). The Euclidean error was higher when the target was absent vs. preent (*Figure 2e*, absent 5.08° vs. present 2.50°, p = 1.4e-91, with FDR) and the same of the distractor (absent 5.25° vs. present 2.60°, p = 6.8e-106, with FDR).

The presence of a person at the gaze goal also impacts the first saccade latency, ($F_{(1,24)}=50.5$, p = 2.4e-07, $\eta^2 = 0.97$). The saccade latency was significantly higher (*Figure 2f*) when a person was absent vs. present at the gaze goal for both target (0.37 s vs. 0.31 s, p = 1.4e-16) and distractor (0.38 s vs. 0.31 s, p = 2.0e-19) trials. There was no difference when the target or distractor person was at the gaze-goal locations for the first saccade error ($F_{(1,24)}=1.94$, p=0.18, $\eta^2 = 0.002$), first saccade

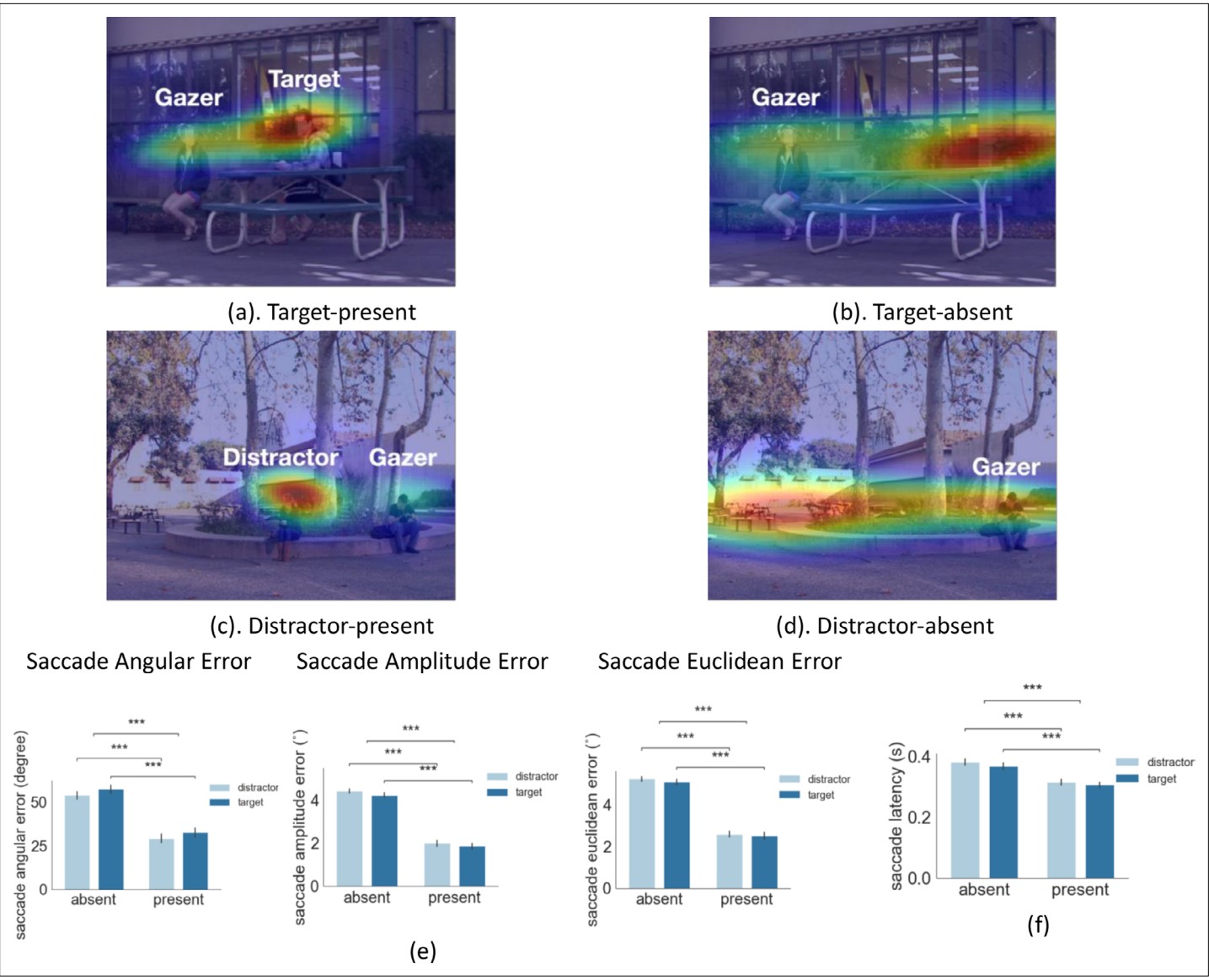

**Figure 2.** Examples of first gaze-following saccade errors and latency. (**a-d**) Examples of first gaze-following saccade endpoint density maps for target-present combining data across all observers. (**e**) First gaze-following saccade endpoint angular error (degrees), amplitude error (degrees of visual angle°), and Euclidean error (degrees of visual angle°) relative to the gaze goal (person's head center). (**f**) First gaze-following saccade latency for target/distractor present or absent at the gaze goal.

angular error (F(1,24)=2.61, p=0.11, $\eta^2 = 0.01$), or first saccade latency (F(1,24)=2.15, p=0.15, $\eta^2 = 0.03$, *Figure 2e–f*).

## Relating eye movement dynamics to gazer information

To relate the dynamics of eye movements (*Figure 3a*) to the gazer's head information throughout the video, we estimated gaze direction using a state-of-the-art deep neural network (DNN) model (*Chong et al., 2020*, *Figure 3a*, see methods for details). The accuracy of the DNN model in estimating the gaze goal location for these images is comparable to that of humans for target/distractor present and superior to humans for target/distractor-absent trials (*Han and Eckstein, 2022*). For each video frame, the model generated a gazer vector in which the start point was the gazer's eye position, and the endpoint was the model-estimated gaze-goal location. From the frame-to-frame gazer vector, we calculated gazer vector distance in degrees of visual angle (°), angular displacement (degree), head velocity (degree/s), and head acceleration (degree/s²) at a sampling rate of 30 frames/s (see

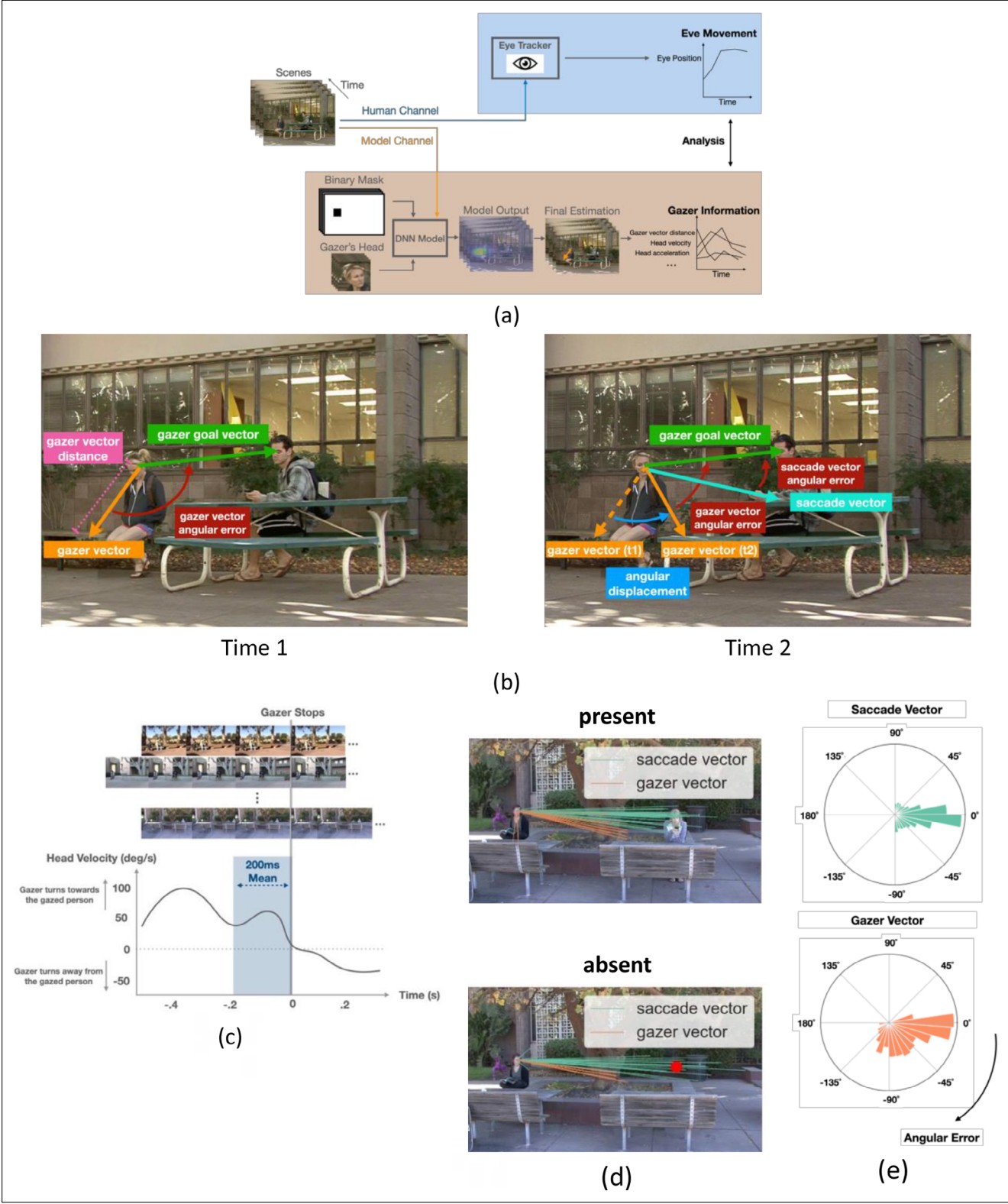

**Figure 3.** AI model analysis, definitions of saccade vectors and gaze vectors, and head velocity over time. (**a**) Workflow for AI model for gaze estimation (*Chong et al., 2020*). The model takes individual frames, paired with a binary mask that indicates the gazer's head location within the scene, and a cropped image of the gazer's head, to produce a probability heatmap. The pixel location with the highest probability was taken as the final estimated gazed location and gazer vector endpoint (orange arrow in final estimation image). We computed various frame-to-frame gaze features based on the gazer vectors and related them to the dynamics of observers' eye movements during gaze-following. (**b**) Examples of the initial gazer vector, the gazer

*Figure 3 continued on next page*

*Figure 3 continued*

vector distance, the gazer goal vector, the angular displacement, and angular errors. The gazer vector distance was the vector length indicating how far away the estimated gazed location (by the gazer) was from the gazer. The gazer goal vector is the vector whose start point was the gazer's head centroid and the endpoint was the gazer goal location. The angular displacement is the angle between the current gazer vector and the initial gazer vector position. The angular error is the angle between the current gazer/saccade vector and the gazer goal vector. (**c**) Estimation of the typical head velocities right before (200ms interval) the gazer's head stops moving. Velocities were obtained by aligning all videos relative to the gaze stop time and averaging the head velocities. Head velocity = 0 at time = 0. (**d**) The first saccade vectors (teal lines) and corresponding gazer vectors (orange lines) at the saccade initiation times for all observers and trials for the same video (top: gaze goal present condition, bottom: gaze goal absent condition). (**e**) Histogram of angular errors for first saccade vectors and gazer vectors at the saccade initiation times for all trials/videos and observers. All vectors were registered relative to the gazer goal vector (the horizontal direction to the right represents 0 angular error).

The online version of this article includes the following figure supplement(s) for figure 3:

**Figure supplement 1.** Histograms of gaze goal estimation standard deviations across human gaze annotations in the horizontal direction (left) and in the vertical direction (right).

Methods for detailed calculation, see *Figure 3b* gaze information definitions). We could then relate the observers' saccade execution times to the moment-to-moment changes in the gazer vector's measures. We also quantified, from the videos, the typical gazer head velocity before the gazer's head stopped. This was accomplished by lining up all videos based on the head stop and averaging the gazer's head velocities (*Figure 3c*).

## Anticipatory first saccades that predict gaze goal direction

The gazers' head movements started with the video onset and their mean duration was 0.61 s. The observers' mean first saccade latency was 0.34 s (std = 0.07 s). Thus, the saccade initiation most often preceded the end of the gazer's head movement. In 81% of the trials, participants initiated the first saccade before the gazer's head movement stopped (86% of the trials for target-present, 85% for distractor-present, and 77% for target/distractor-absent trials). We investigated whether these anticipatory first saccades were based on a prediction beyond the available information in the gazer's head direction at the time of saccade initiation. Or on the contrary, are the saccade directions based on the information in the gazer's head direction at the time of saccade initiation?

To evaluate these hypotheses, we measured the angular error between the DNN-estimated gazer's head direction (gazer vector) at the time of the first saccade initiation and the gazer goal vector (*Figure 3b* right) for each trial. The gazer vector angular error at the time of saccade initiation provides a lower bound on observers' saccade angular error if the brain only used the gazer's head direction to program the eye movements. *Figure 3d* visualizes the first saccade vectors (teal lines) and corresponding gazer vectors (orange lines) at the saccade initiation times for all observers and trials for a sample video. The results show how the saccade directions are closer to the gazer goal direction than the direction information provided by the gazer's head at the time of saccade initiation (gazer vector). *Figure 3e* shows co-registered saccade vectors and gazer vectors at the time of saccade initiation across all trials/observers. The horizontal line pointing to the right represents zero angular error (i.e. a saccade or gazer vector that points in the same direction as the direction of the gazer goal). The mean angular error for the saccade directions was significantly smaller than that of the gazer vector at the time of saccade initiation (18 degrees vs. 40 degrees, bootstrap p<1e-4, Cohen's d=0.71). This difference was larger for target/distractor-present videos (14 degrees vs. 42 degrees, bootstrap p<1e-4, Cohen's d=0.34) but was still significant even when the target/distractor was absent (22 degrees vs. 38 degrees, bootstrap p<1e-4). The findings suggest that observers make anticipatory first saccades that infer the direction of the gaze goal beyond the momentary information from the gazer's head direction. We estimated the additional time after saccade initiation it took for the gazer's head to point in the direction of the saccade. On average it took 0.37 s (std across observers = 0.09 s) and 0.22 s (std across observers = 0.09 s) for the gaze vector to reach the saccade vector direction for videos with target/distractor-present and target/distractor-absent respectively.

To make sure the results were not due to inadequate gaze estimates by the DNN, we repeated the analysis with humans-estimated gazer vectors instead of DNN-estimated gazer vectors. The human-estimated gazer vectors were obtained from ten individuals (not participants in the study) that viewed randomly sampled individual frames from the videos and were instructed to select the gaze goal (see methods). Because we were interested in measuring the inherent information provided by the gazer's

head direction independent of the peripheral information, the participants viewed frames from the target/distractor-absent videos. The human-estimated gazer vectors resulted in smaller angular errors than the DNN but showed similar findings. Observers' mean first saccade angular error was significantly smaller than the mean human gazer vector angular error (18 degrees vs. 32 degrees, bootstrap p<1e-4). This effect was present for both, the target/distractor-present videos (14 degrees vs. 36 degrees, bootstrap p<1e-4, Cohen's d=0.56) as well as the target/distractor-absent (22 degrees vs. 27 degrees, bootstrap p=0.017, Cohen's d=0.11). On average it took 0.34 s (std = 0.12 s) and 0.16 s (std = 0.09 s) for the gazer vector to reach the 1st saccade vector location for the present and absent conditions.

## Frequent reverse saccades triggered by low velocity in the gazer's head rotation

Even if we explicitly instructed participants to follow the gaze, our analysis of eye position revealed that participants executed backward saccades in the opposite direction of the gazer vector (reverse saccades) in 22% of all trials (see *Figure 4a* , demo video *Video 1*). The mean reverse saccade initiation time was 0.63 s (std = 0.07 s, *Figure 4b*) with a mean amplitude of 3.5° (std = 1.2°). Over 80% of the reverse saccades were either the second or the third saccade in the trial (reverse saccade index, *Figure 4b*). The mean duration of the gazer's head movement during reverse saccade trials was 0.65 s. In 87% of the videos, the gazer started to look away from the gazer person at the end of the movie (DNN estimation mean = 0.98 s, std = 0.18 s, human estimation mean = 1.06 s, std = 0.15 s). In those videos, the majority of reverse saccades (88%) were executed before the gazer started looking away. *Figure 4c* shows the frequency of first saccades and reverse saccades, as well as the overall head velocity over time. Trials with reverse saccade had significantly shorter first saccade latencies compared to those without reverse saccade (F(1,24)=96.8, p = 6.8e-10, $\eta^2$ = 0.84, *Figure 4d*, target/distractor-present condition 0.23 s vs. 0.34 s, p=2.6e-67, absent condition 0.27 s vs. 0.40 s, p=4.2e-50, both posthoc pairwise comparison with FDR). What could explain the shorter first saccade latencies of trials with reverse saccades? One possible interpretation is that early first saccades are unrelated to the stimulus properties and are generated by stochastic processes internal to the observer. Consequently, when the first saccade is executed too early, a compensatory reverse saccade is subsequently programmed.

An alternative possibility is that the observer's early first saccade executions are not random but related to some aspect of the gazer's head movement. To investigate this possibility, we first analyzed the average head velocity over time relative to the timings of the video onset (coincident with the gazer head movement onset) and first saccade execution. The analysis was done separately for trials with and without reverse saccade. If the early first saccades in reverse saccade trials are triggered randomly and are unrelated to the gazer's head features, we should find no significant difference in average head velocity between the two types of trials. Instead, we found a significantly lower head velocity during the first 0.23 s of the video for the trials with reverse saccades, 63.6 degrees/s vs. 93.6 degrees/s, cluster-based permutation test, p=1.0e-04, Cohen's d=1.9 (*Figure 4e*, average head velocity lined up with video start). When we aligned the data with the initiation time of the first saccade, we also observed a significantly lower head velocity for the trials with reverse saccade during 0.37 s before the first saccade initiation 47.1 degrees/s vs. 104.9 degrees/s, cluster-based permutation, p = 1.0e-4, Cohen's d=2.3 (*Figure 4f*). Furthermore, the average head velocity of 47.1 degrees/s was within the range [31.6 degrees/s - 62.5 degrees/s, 95% confidence interval] of the average head velocity before the gazer's head stops (estimated from all movies; see the horizontal green band in *Figure 4e and f*). These findings suggest that when the gazer's head velocity is slow, observers make an inference that the gazer might be stopping her or his head movement. Observers then execute an eye movement to the currently estimated gazer goal. Thus, observers' faster first saccades are not executed at random times but are related to the observers' inference that the gazer's head movements might come to a stop. *Figure 4f* also shows that right before the execution of the first saccade, in reverse saccade trials, the head velocity starts accelerating. We interpret this to indicate that observers infer from the accelerating velocity just prior to the first saccade execution that the gazer's head will not come to a stop. Consequentially, observers program a reverse saccade.

Our analysis focused on the head velocity, but what about other features of the gazer's head? The *Figure 4—figure supplement 1* shows analyses for other features including distance, angular

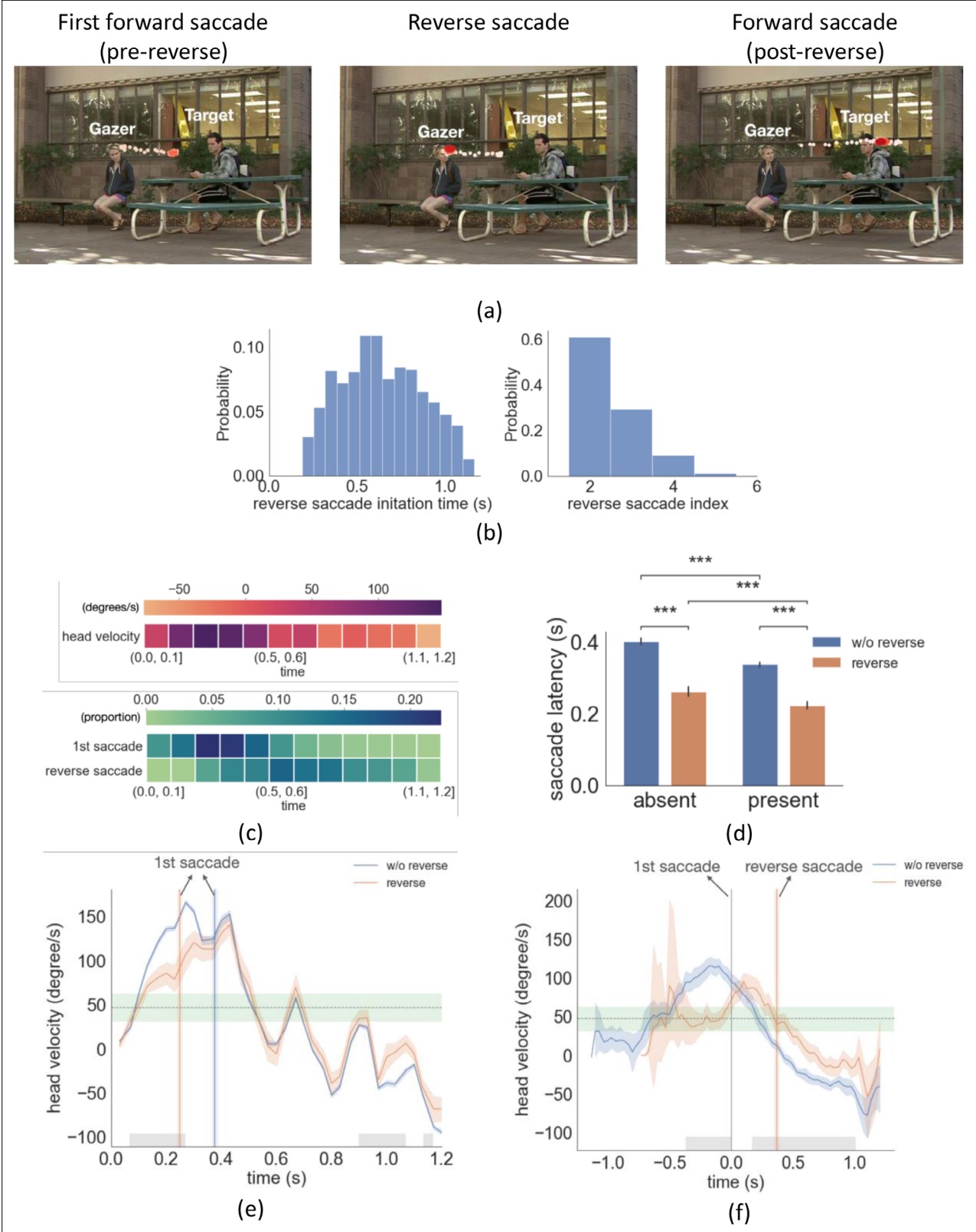

**Figure 4.** Example trial with reverse saccade, distributions of saccade time and latency, head velocity difference between trials with reverse saccades and trials without reverse saccades. (**a**) An example of eye movement trace for three saccades over time. A first gaze-following saccade, followed by a reverse saccade, and another post-reverse saccade gaze-following saccade. The light red to dark red represents the order in time (see video demo at https://osf.io/yd2nc). (**b**) Histogram of the reverse saccade initiation time and reverse saccade index (2nd, 3rd, etc.). (**c**) Heatmaps represent the first

*Figure 4 continued on next page*

*Figure 4 continued*

saccade and reverse saccade frequency, and the gazer's head velocity over time (**d**) Saccade latency separated by three conditions and reverse saccade trials. (**e**) Gazer's head rotation velocity vs. time separated for reverse saccade and non-reverse saccade trials. Shaded areas are the 95% bootstrapped confidence interval. Positive velocity represents the head moving toward the gazed person's location. The vertical lines are the mean first saccade latency. The gray area shows the statistical significance under the cluster-based permutation test. The green area represents the 95% confidence interval of the velocity right before the gazer's head stops moving across all movies. (**f**) The same figure as (**e**) except that head velocity was aligned at the initiation time of the first saccade. (**g**) The proportion correct for linear support vectore machine (SVM)models trained to predict whether a movie was in the upper 50 %/or lower 50% of movies based on the number ofreverse saccades. The x-axis is the time range from the movie used to train the SVM model. The first saccade latency and reverse saccade latencies are marked as dashed lines as references. (**h**) The head velocity aligned with the first saccade initiation time at t=0, separately for trials with frozen frames and without.

The online version of this article includes the following figure supplement(s) for figure 4:

**Figure supplement 1.** Gazer vector distance, angular displacement, and head acceleration over time, averaged across videos with t=0 aligned with video onset (left column) and t=0 aligned with first saccade onset (right column).

displacement, and head acceleration. Other features are also significantly different across reverse and non-reverse saccade trials. This is not surprising because there is a correlation between some of the features. For example, before the first saccade execution, the angular displacement is smaller for reverse saccade trials. This is because slower angular velocity for the head will result in lower angular displacement at the time before the first saccade. Still, head velocity showed the clearest results. To further investigate whether the head velocity or other gaze features from the videos can better explain reverse saccades, we trained multiple support vector machine (SVM) models using different head features to predict the frequency of reverse saccades (binary prediction: top vs. bottom 50 percentile) using features: 1. Gazer vector distance 2. Angular displacement, 3. Head angular velocity, 4. Head angular acceleration (see Methods for detailed description). We used the time range starting from the beginning of the video and gradually increased the time range for the predictor, and plotted the SVM model proportion correct (PC) in *Figure 4g*. We found that the head velocity had the highest accuracy in predicting reverse saccade movies among all gaze features. The model's accuracy peaked when we used head velocity information from 0 to 230ms of each video (71.2%) and asymptoted afterward. This was consistent with the results that during the first 0.23 s of movies, trials with reverse saccade had a significantly lower head velocity than those without.

## When are the reverse saccades planned?

Having established that the gazer's low head velocity might be triggering an early first saccade in trials with reverse saccades, we tried to determine the timing of the reverse saccade programming. One possibility is that the reverse saccades are programmed after the execution of the first forward saccade. In this framework, the gazer's initial slow head velocities in some trials trigger an early first saccade forward and, during that subsequent fixation, the motion of the gazer's head accelerating captures attention and triggers the reverse saccade. A second possibility is that it is the gazer's head velocity increase right before the observer executes the first saccade (*Figure 4f*) that triggers the programming of the reverse saccade prior to the execution of the first forward saccade. To assess these two hypotheses, we conducted another experiment, in which we monitored in real time the eye position of observers and froze the video frames immediately after participants initiated the first gaze-following saccade (*Figure 4h*, demo video *Video 2*). This only occurred randomly in 50% of the trials to prevent observers from changing their eye movement strategy. If observers' reverse saccades were triggered by the transient motion after the first saccade execution, then freezing the video and eliminating the transient peripheral motion signal of the head should diminish the frequency of the reverse saccades. However, we found that freezing the video frame after the first saccade execution did not reduce the proportion of trials with reverse saccade relative to the

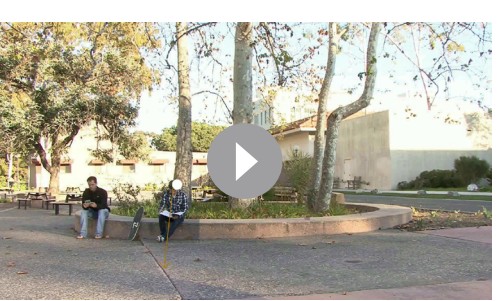

**Video 1.** Example trial with reverse saccade (white dot represents gaze position, yellow arrow represents gaze vector).

https://elifesciences.org/articles/83187/figures#video1

unfrozen videos trials, (mean = 22%, std = 12% for frozen vs. mean = 21%, std = 11% for unfrozen, bootstrap p=0.6). These results suggest that observers planned the reverse saccade prior to the execution of the first forward saccade.

## Functional role of reverse saccade

Next, we tried to understand the function, if any, of reverse saccades. We first analyzed the endpoint of the reverse saccade. We found that the reverse saccades landed close to the gazer's head (*Figure 5a*; mean distance to the gazer's head 0.79°, std = 0.28°) suggesting that the reverse saccades aim to re-fixate the gazer given the change in the gazer vector after execution of the first saccade. Thus, the reverse saccades could be more precisely described as 'return saccades'. To assess the potential functionality of the reverse saccade, we compared the error in fixating the gaze goal (saccade error: saccade endpoint distance to the gazed person's location) of forward saccades before and after the reverse saccade. *Figure 5b* shows the density map of forward saccade endpoints separately for pre and post-reverse saccades for a single sample image, as well as the density map combined across all images by registering the saccade endpoints relative to the gazer's head. We found that saccade angular error was lower following the reverse saccades for both the target/distractor-present condition (*Figure 5c*, 20.53 degrees for post-reverse saccade vs. 33.7 degrees for pre-reverse, bootstrap p=0.0014, Cohen's d=0.44, corrected by FDR) and for the target/distractor-absent condition (34.37 degrees for post-reverse saccade vs. 71.01 degrees for pre-reverse, bootstrap p<1e-4, Cohen's d=1.2, corrected by FDR).

The saccade amplitude error was lower following reverse saccades (*Figure 5d*; pre-reverse saccade1.6° vs. post-reverse saccade 1.3°, p=0.017, Cohen's d=0.38, based on bootstrap resampling, see methods). For the target/distractor-absent condition, we did not find this effect, pre- 4.0° vs. post- 4.3°, p=0.06, Cohen's d=0.29. Overall, forward saccades following a reverse saccade ended closer to the gaze goal than the saccades before reverse saccades (Euclidean error, *Figure 5e*; 1.8° for post-reverse saccade vs. 2.5° for pre-reverse, p=0.0054, Cohen's d=0.5). For the target/distractor-absent condition, the effect did not reach significance, 5.2° vs. 5.3°, p=0.43, Cohen's d=0.07.

Finally, the saccade Euclidean error in the trials without reverse saccades was significantly lower compared to the trials with reverse saccades (distractor/target-present w/o reverse saccade 1.1° vs. with reverse saccade 1.8°, Cohen's d=0.79; target/distractor-absent w/o reverse saccade 3.7° vs. with reverse saccade 5.2°, Cohen's d=1.4, all p<0.001, corrected by FDR, *Figure 5e*). This result suggests that the gazer information was less ambiguous and more accessible to observers in the trials with no reverse saccades.

## Causal influence of re-foveating the gazer with a reverse saccade

Our analysis showed that the saccade endpoint after the reverse saccade was closer to the gaze goal (and smaller angular error) than the endpoint of the forward saccade preceding the reverse saccade. The interpretation is that re-fixating the gazer with the reverse saccade improved the inference about the gazer goal and benefited the subsequent forward saccade. However, an alternative explanation is that the gaze-following saccade after a reverse saccade simply has longer visual processing compared to the first saccades preceding the reverse saccades (first saccade initiation time m=0.35 s vs. first saccade post-reverse saccade initiation time m=0.84 s). Longer processing times would result in better estimates of the gaze goal.

To assess these two competing explanations for the reduction of error of gaze-following saccades after a reverse saccade, we implemented a follow-up experiment with twenty-five new observers. In the new experiment, we digitally erased the gazer on 50% of the reverse saccade trials (randomly) before the re-fixation of the gazer. To accomplish this, we monitored eye position in real-time, and whenever we detected a reverse saccade during the video, we erased the gazer with a 50% probability (demo video *Video 3*). The experiment allowed comparing the errors of gaze-following saccades after a reverse saccade with matched visual processing times. If the reduced saccade error is related to the foveal re-processing of the gazer after the reverse saccade, we should expect a larger saccade error when we erase the gazer (see *Figure 6a* example).

We first confirmed that the basic analyses replicated the first experiment. The mean reverse saccade initiation time was 0.69 s (std = 0.07 s), with 80% of the reverse saccades being the second or the third saccade. Reverse saccades occurred in 31% of the trials. Trials with reverse saccade had a

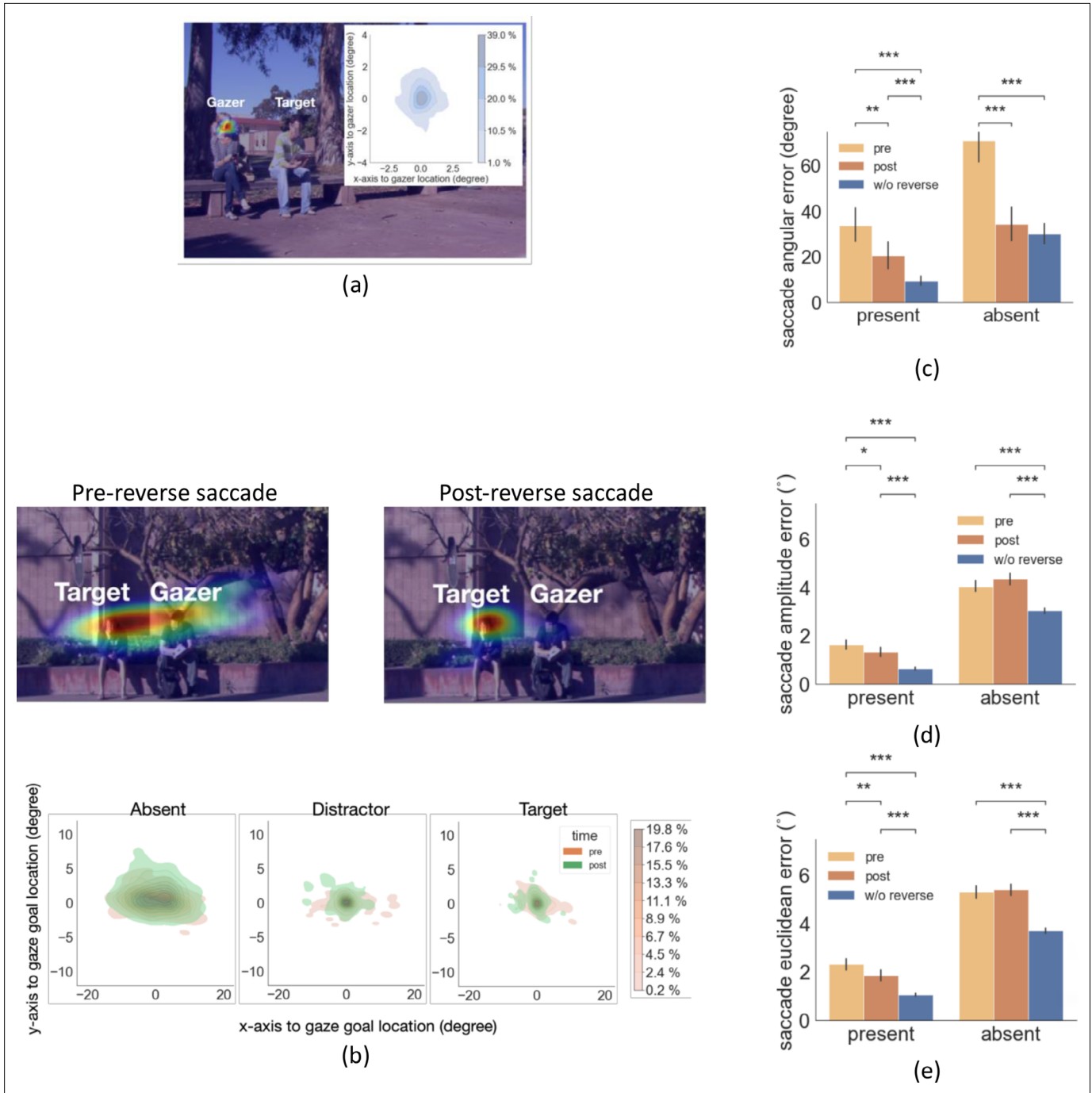

**Figure 5.** Distribution of reverse saccade landing positions and saccade errors. (**a**) Density map of reverse saccade endpoint locations overlayed on an example image. The density map of all reverse saccade locations registered across videos relative to the gazer's head location at origin (0,0) is shown on the top right. Colorbar shows the proportion of saccades falling in each region. (**b**) Top: Density map of gaze-following saccade location pre- and post-reverse saccade overlaying on an example image. Bottom: Density map of all saccades pre- and post-reverse saccades registered relative to the gazed person's head locaton at origin (0,0). Colorbar shows the proportion of saccades falling in each region. (**c-e**) The saccade angular error (angular difference between the saccade vector relative to the gaze goal vector), the saccade amplitude error (amplitude difference between the saccade vector relative to the gaze goal vector), and the saccade Euclidean error (relative to the gazed location, center of the head) for pre- and post-reverse saccades. Trials with no reverse saccade were treated as the baseline conditon.

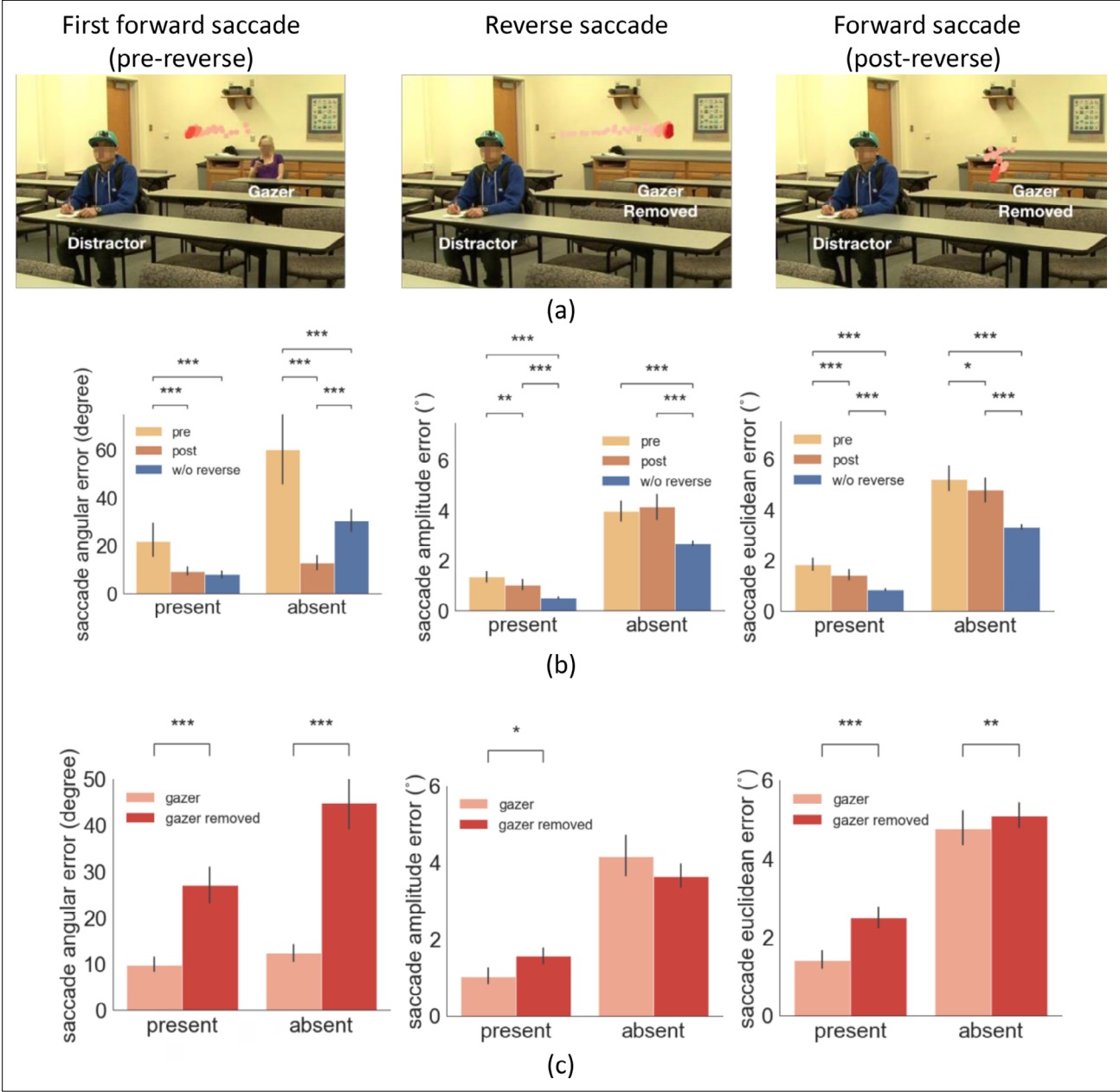

**Figure 6.** Example trial of reverse saccade when the gazer was removed and saccade errors. (**a**) Example of eye movement trace over time when the gazer was erased triggered by the detection of a reverse saccade. The light red to dark red represents the order in time (see video demo at https://osf.io/etqbw). (**b**) The saccade angular error (angular difference between saccade vector relative to gaze goal vector), the saccade amplitude error (amplitude difference between saccade vector relative to gaze goal vector), and the saccade Euclidean error (saccade endpoint location relative to gazed person's head) in trials without gazer removed pre-reverse saccade vs. post-reverse saccade vs. baseline trials (w/o reverse saccades). (**c**) The saccade angular error, saccade amplitude error, and the saccade Euclidean error post reverse saccade with gazer removed vs. gazer unaltered.

The online version of this article includes the following figure supplement(s) for figure 6:

**Figure supplement 1.** Gazer's head velocity averaged across all videos aligned (t=0) relative to the start of the videos (**a**) and aligned relative to the first saccade initiation time (**b**) for experiment 2.

**Figure supplement 2.** The proportion of trials that have the first saccade moving towards the gaze goal for the absent and the present condition in the free-viewing experiment.

significantly smaller first saccade latency compared to those without reverse saccade (0.23 s vs. 0.33 s, bootstrap p<1e-4, Cohen's d=1.2). Reverse saccade trials were associated with slower head velocity during the initial period of the movie (100 ms-260ms) and 150ms before the first saccade (*Figure 6—figure supplement 1*). For trials without the gazer removed, we found similar pre- vs. post- reverse saccade error results as in the first experiment, saccade angular error (target/distractor-present pre- 19.4 degrees vs. post- 6.7 degrees, bootstrap p<1e-4, Cohen's d=0.4; target/distractor-absent, pre- 53.9 degrees vs. post- 12.7 degrees, bootstrap p<1e-4, Cohen's d=1.3), saccade amplitude error (target/distractor-present pre- 1.5° vs. post- 1.0°, bootstrap p=0.0014, Cohen's d=0.28; but not for target/distractor-absent pre- 4.10° vs. post- 3.8°, bootstrap *P*=0.25, Cohen's d=0.17), and saccade Euclidean error (target/distractor-present pre- 1.8° vs. post- 1.4°, bootstrap p=1.2e-4, Cohen's d=0.3; target/distractor-absent pre- 5.2° vs. post- 4.8°, bootstrap p=0.04, Cohen's d=0.4) (*Figure 6b*).

Critical to our hypotheses of interest, the results showed that the saccade angular error post-reverse saccade was significantly higher in the trials with the gazer removed compared to those with unaltered videos, for both the target/distractor-present condition (28.2 degrees pre-reverse saccade vs. 9.2 degrees, post-reverse saccade, bootstrap p<1e-4, Cohen's d=1.05) and absent conditions (46.0 degrees pre-reverse saccade vs. 12.8 degrees post-reverse saccade, bootstrap p=1e-4, Cohen's d=1.06, *Figure 6c*). Significantly higher saccade amplitude error with removed gazer was only found for the target/distractor-present condition (1.5° vs. 1.0°, bootstrap p=0.0136, Cohen's d=0.34) but not for absent conditions (3.6° vs. 4.1°, bootstrap p=0.37, Cohen's d=0.09, *Figure 6c*). Finally, the saccade Euclidean error post-reverse saccade was significantly higher in the trials with the gazer removed for both the target/distractor-present condition (2.5° vs. 1.4°, bootstrap p<1e-4, Cohen's d=0.7) and the absent condition (5.1° vs. 4.8°, bootstrap p=0.006, Cohen's d=0.6; *Figure 6c*). The time of the forward saccade following the reverse saccade was the same across trials with the gazer removed or unaltered (0.83 s from video onset with the gazer unaltered vs. 0.8 s, with the gazer removed, bootstrap p=0.1). This finding confirms that the benefit of reducing the gaze-following saccade errors is causally linked to the uptake of additional gaze goal information from re-fixating the gazer with a reverse saccade.

## Anticipatory and reverse saccades during free-viewing search

In our two search experiments, we instructed observers to follow the gaze of the person in the video. This specific instruction might be unnatural and might have motivated observers to follow the gazer's head movements and trigger anticipatory saccades and reverse saccades. To assess the generality of our findings we implemented a control experiment (Experiment 3) with five participants (see Methods for sample size power estimation) where we did not explicitly instruct observers to follow the gaze during the video presentation. Instead, we only instructed them to evaluate whether they could find the target person and decide whether they were present (yes/no task, 30% prevalence). No information was given about the gazer or eye movement strategies to follow. We found that participants spontaneously executed gaze-following saccades for 74% (std = 10%) and 91% (std=7.4%) of the trials for the absent and the present condition, respectively (*Figure 6—figure supplement 2*). Observers also executed anticipatory first saccades prior to the end of the gazer's head movement in 88% of the trials. We also observed an even larger number of reverse saccades than in the first experiment where observers were instructed to follow the gazer (33%, std = 22% and 37%, std = 16%) of trials for the absent and the target or distractor present condition, respectively. These findings suggest that anticipatory and reverse saccades are not a byproduct of the instructions in experiment 1.

## Discussion

We investigated eye movement control while following the gaze of others. Although human eye movements are fast and might seem idiosyncratic, our findings show that the human brain uses moment-to-moment information about the gazer's head dynamics and peripheral information about likely gaze goals to rationally plan the timing and endpoint of saccadic eye movements.

First, we found that the oculomotor system integrates information about the foveally presented gazer's head and peripheral information about potential gaze goals. When a person was present at the gaze goal, observers executed faster and more accurate saccades. One might have expected the presence of gaze goals to reduce the saccade endpoint Euclidean error but to a lesser degree reduce saccade angular error. Arguably, if observers waited for the gazer's head to stop, they could rely on

the head direction to accurately plan their gaze-following saccade direction without needing to rely on a peripheral gaze goal. Our results showed otherwise. Deleting the gaze goal also increased the saccade angular error. This suggests that the eye movement direction also heavily relies on the peripheral processing of likely gaze goals. This might seem counterintuitive but if one considers that the saccades are programmed before the gazer's head comes to a stop and points to the gaze goal, then the gazer's head direction might not provide sufficient information for observers to program saccades to the correct direction.

The evidence for the integration of foveal and peripheral information is consistent with a series of studies showing observers' ability to simultaneously process foveal and peripheral information for simpler dual tasks with simple stimuli (*Ludwig et al., 2014*; *Stewart et al., 2020*) and their joint influence on fixation duration during scene viewing (*Laubrock et al., 2013*) and subsequent eye movements (*Wolf et al., 2022*).

Importantly, we found the first saccades are anticipatorily initiated before the gazer's head movement comes to a stop. And they contain information about the direction of the gaze goal that is more accurate than the direction information provided by the gazer's head at the time of saccade initiation. This suggests that the brain is using peripheral information to make an active prediction about likely gaze goals. We found that on average the first saccades were anticipating the head direction by 340–370ms and 160–220ms for target present and absent conditions. Furthermore, previous studies have shown that a saccade is typically based on visual information presented ~100ms before saccade execution (*Becker and Jürgens, 1979*; *Caspi et al., 2004*; *Hooge and Erkelens, 1999*; *Ludwig et al., 2005*). Thus, the first saccade might only have access to the gazer's head direction up to ~100ms before saccade execution, which means the observes' first saccades might be anticipating the gazer's head direction by 440–470ms and 260–320ms. Similarly, the gazer's head direction at the time of saccade programming (rather than execution) was, on average, pointing 44.1 degrees (for target/distractor present trials) and 39.7 degrees (for target/distractor absent trials) away from the gaze goal.

The evidence for an inferential process that influences saccade programming when a person is present as the gaze goal in the scene might be expected. But, the inferential process still prevailed when a target/distractor was absent. It is likely that even when no person is present at the gaze goal location, the brain uses information about the scene including the ground, the objects, and the sky to make estimates of likely gaze goals. Prior knowledge about the maximum angular rotation of the gazer's head also constrains the possible gazer goals. This information is used by observers to program anticipatory inferential eye movements even in the absence of an unambiguous visible gazer goal (e.g., a person) in the observer's visual periphery.

Second, we found that early first saccades are executed when the gazer's head velocity diminishes to values comparable to the velocity that is typical during the 200ms time interval before the head stops. This is consistent with the idea that observers use the gazer's head velocity to dynamically make inferences about the likelihood that the head will stop and then execute a saccade towards the likely gaze goals. However, our data also suggest that other cues are used to infer that the gazer's head will stop. For example, for some trials with longer first saccade latencies (no reverse saccade trials), the head velocity before the observers' saccade execution is almost double the typical head velocities during the 200ms time interval before a head stops (*Figure 4e*). Thus, the observers must rely on other cues. In these long latency trials (*Figure 4e*) there is a reduction of the head velocity in the 200ms before the saccade execution suggesting that observers use the head's deceleration to infer that the gazer's head will come to a stop and then execute the first gaze-following saccade. It is also likely that for trials with a gaze goal, observers use an estimated error between the implied gaze direction and the gaze goal to plan saccades. Small estimated angular errors might be used to trigger saccades. Thus, we suggest that the observers' oculomotor system might use multiple cues (head velocity, head deceleration, estimated gaze errors, etc) to infer that the gazer's head might stop and trigger gaze-following saccades.

Third, surprisingly, we found that observers often executed reverse saccades in a significant proportion of trials (>20%). These reverse saccades are directed to re-fixate the gazer's head and can be referred to as return saccades. The reverse saccades were not an artifact of our instruction to the observers to follow the gaze of the person in the video. A follow-up experiment where observers were instructed to decide whether a target person was present with no instruction to participants

about their eye movements also resulted in a comparable proportion of reverse saccades. Why might observers make such saccades? Our analysis showed that these reverse saccades do not appear randomly across trials. Reverse saccades occur on trials in which the gazer's head velocity is slow but starts accelerating about 200ms before the first saccade is executed and observers infer that the gazer's head will not come to a halt. Why don't observers simply cancel the forward saccade? Studies have shown that there is a 50–100ms delay between the programming of a saccade and its execution (*Becker and Jürgens, 1979*; *Caspi et al., 2004*; *Ludwig et al., 2005*). The gazer's head acceleration occurring immediately before the execution of the forward saccade is not used to cancel the impending planned eye movement.

Our findings with the experiment that freezes the gazer after the first forward saccade suggest that the reverse saccade is programmed before the execution of the first forward saccade. This concurrent programming of saccades has been documented for simplified lab experiments (*Becker and Jürgens, 1979*; *Caspi et al., 2004*) but not in the context of real-world stimuli and tasks. One alternative explanation we did not explore is that reverse saccades are simply triggered after the first forward saccades that do not land on the target/distractor. In this perspective, a forward saccade is executed and when foveal processing determines that the saccade endpoint was far from a likely gaze goal then a reverse saccade is programmed and executed (regardless of the velocity of the gazer's head). Data analysis does not favor this interpretation. In a small percentage of trials (15% of reverse saccade trials) first saccades landed within 0.5° visual angle of the target but these were still followed by reverse saccades. This observation suggests that foveating close to the gaze goal was not sufficient to interrupt a reverse saccade programmed before the execution of the first forward saccade.

Our results also show that the reverse saccades had functional importance. Forward saccades, after re-fixating the gazer with a reverse saccade, were more accurate at landing close to the gaze goal. The benefit of re-fixating the gazer was more reliable when there was a person present at the gaze goal. When a gaze goal person was absent we found a less reliable re-fixation benefit (not statistically significant in experiment 1 and marginally significant in experiment 2) suggesting that not having a peripheral likely gaze goal can be a bottleneck to the accuracy of saccade endpoints.

The existence of reverse saccades to re-fixate the gazer's head might seem puzzling. Why does the oculomotor programming system not wait longer until the gazer's head comes to a full stop, then executes the gaze-following saccade and avoids programming reverse saccades altogether? Executing anticipatory eye movements that predict future grasping actions (*Mennie et al., 2007*; *Pelz and Canosa, 2001*), the location or motion of a stimulus (*Fooken and Spering, 2020*; *Kowler, 1989*; *Kowler, 2011*; *Kowler et al., 2019*) is common for the oculomotor system. Thus, while following the gaze of others the observers' oculomotor system plans anticipatory saccades that predict the gaze goal before the completion of the gazer's head movements. Occasionally, these predictive saccades are premature and the brain rapidly programs a reverse saccade to re-fixate the gazer and collect further information about the potential gazer goal.

Are the reverse saccades unique to gaze following? No, humans make reverse saccades in other visual tasks that require maintaining information in working memory, such as copying a color block pattern across two locations (*Hayhoe and Ballard, 2005*; *Hayhoe, 2017*; *Meghanathan et al., 2019*). Most notably during reading humans make frequent reverse saccades ("called regressive"). Although one might draw a parallel between reading and gaze-following, our findings highlight important distinctions. Regressive saccades during reading are related to inaccurate eye movements that missed critical words or fixations that are too short to deeply process a word's meaning (*Inhoff et al., 2019*; *Rayner, 1998*). The reverse saccades while following dynamic gaze are related to moment-to-moment changes in the visual information in the world (i.e. the gazer's head velocity) and the oculomotor systems' rapid response to optimize gaze-following.

A remaining question is whether the gaze-following behaviors documented in the current study reflect real-world eye movement strategies. Do the reported inferential anticipatory and reverse saccades generalize to the real-word or are they a consequence of the timing of our study or the particular tasks in the study? The videos were presented for 1.2 s which should not represent an excessive time pressure to observers when compared to other eye movement studies on faces (*Peterson and Eckstein, 2012*) or search (*Ackermann and Landy, 2013*; *Eckstein et al., 2015*). We evaluated a person search task with explicit instructions to follow the gazer and a second task with no eye movement instructions and observed reverse and anticipatory saccades for both tasks. There is also

evidence that for natural tasks such as looking at faces or searching, many of the findings with images in the laboratory do generalize to real word settings (*Mack and Eckstein, 2011*; *Peterson et al., 2016*). Still, the generalization to the real world for gaze-following eye movement behaviors needs to be assessed with mobile eye trackers (*Land and Hayhoe, 2001*).

What might be the brain areas involved in the oculomotor programs for gaze following? There is a large literature relating gaze position to neuronal response properties in the superior temporal sulcus (*Oram and Perrett, 1994*) and dorsal prefrontal cortex (*Lanzilotto et al., 2017*). These areas relay information to the attention and gaze network in the parietal and frontal cortex which are responsible for covert attention and eye movements (*Pierrot-Deseilligny et al., 2004*). Finally, the concurrent programming of saccades has been related to neurons in the Frontal Eye Fields (FEF, *Basu and Murthy, 2020*). Identifying brain areas that integrate peripheral information to generate predictions of likely gaze goals is an important future goal of research.

There are various limitations of the study. In our study, the target/distractor person always appeared at the gaze goal. Thus, the peripheral presence of a person provided some certainty to observers that such a person would be the gazer's goal. If we had shown videos in which gazers looked at one of two people or other objects even when a person was present, the larger uncertainty about the gazer goals might delay the execution of observer saccades. Thus, cognitive expectations which have been shown to play an important role in oculomotor control (*Kowler, 1989*) will influence our findings. Our studies used people as gaze goals which are quite visible in the visual periphery. Gaze goals less visible in the observers' periphery will influence the accuracy and timing of the saccades (*van Beers, 2007*; *Han et al., 2021a*). In addition, observers might use prior expectations of what types of targets are often looked at (people vs. the floor) to bias their inferences. There is also literature comparing how the human attention system varies when the gaze goals are objects vs. social entities (*Mares et al., 2016*). Thus, the social nature of the gaze goal, as well as social variables about the gazer, might also influence the oculomotor dynamics investigated (*Dalmaso et al., 2020*; *Macdonald and Tatler, 2013*).

Finally, our study focused on the head movement while a large literature focuses on the influences of the gazer's eyes (*Driver et al., 1999*; *Friesen et al., 2004*; *Langton et al., 2000*; *Mansfield et al., 2003*; *Ristic et al., 2002*). Our study was relevant to gazers situated at a distance from the observers. The mean angle subtended by the heads in our videos (1.47°, std = 0.32°) would match the angle subtended by a real-sized head viewed at a distance of 9.3 m (std = 2.0 m) in real life. At that distance, the eye subtends a mean angle of 0.147° (vertically) providing a poor source of information to infer the gaze goals compared to the head orientation. Future studies should investigate gazers at smaller distances from the observer and assess how dynamic gazer eye and head movements are integrated and their interactions (for static images see *Balsdon and Clifford, 2018*; *Cline, 1967*; *Langton, 2000*; *Langton et al., 2004*; *Otsuka et al., 2014*). Similarly, we did not analyze lower body movements. Recent studies have shown the diminished influence of the lower body on the orienting of attention (*Han et al., 2021b*; *Pi et al., 2020*).

To conclude, our findings reveal the fine-grained dynamics of eye movements while following gaze and the inferential processes the brain uses to predict gaze goals and rapidly program saccades that anticipate the information provided by the gazer's head direction. Given that attending to the gaze of others is an integral part of a normal functioning social attention system, our findings might provide new granular analyses of eye movement control to assess groups with social attention deficits for which simpler gaze-following analyses have shown disparate results (*Chawarska et al., 2003*; *Nation and Penny, 2008*; *Ristic et al., 2005*).

## Materials and methods
### Experiment 1
#### Subjects
Experiment protocols were approved by the University of California Internal Review Board. Twenty-five undergraduate students (ages 18–20, 16 females, 9 males) from the University of California Santa Barbara were recruited as subjects for credits in this experiment. All have normal to corrected-to-normal vision. All participants signed consent forms to participate in the study.

## Experimental setup and stimuli

All videos were presented at the center of a Barco MDRC 1119 monitor with 1280×1,024 resolution, subtending a visual angle of 18.4° x 13.8° (width x height). Participants' eyes were 75 cm from the computer screen with the head positioned on a chin rest while watching the videos (0.023° visual angle/pixel). Each subject's left eye was tracked by a video-based eye tracker (SR Research Eyelink 1000 plus Desktop Mount) with a sampling rate of 1000 Hz. Subjects' eye movements were calibrated and validated before the experiment. Any large eye drifts that caused failure in maintaining fixation at the beginning of each trial (>1.5° visual angle) would result in observers having to do a recalibration and revalidation. Events in which velocity was higher than 35°/s and acceleration exceeded 9500°/s² were recorded as saccades.

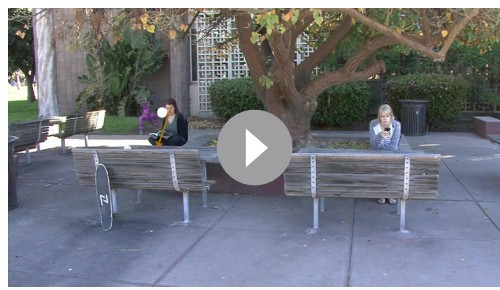

**Video 3.** Example trial when the gazer was frozen after the first gaze-following saccade (white dot represents gaze position, yellow arrow represents gaze vector).
https://elifesciences.org/articles/83187/figures#video3

Stimuli consisted of 80 videos (1.2 s long) originally taken from videos recorded at the University of California Santa Barbara campus in different settings (classrooms, campus outdoors, student apartments, etc.). During the filming, we gave verbal instructions to the actor to look toward another person. Once the video starts, one gazer initiated the gazing behavior (looking at another person) toward either a distractor person (50% chance) or a designated target person (50% chance). A target person is defined as a person that observers needed to search for during the task. There is one target person (male) for the entire experiment. Distractors were defined as all other people that were not the target person. Videos were filmed across different days so that the target person appeared in different clothing across videos/trials. The mean eccentricity of the gazed person relative to the gazer was 6°, std = 3°, median eccentricity was 4.93°, with a minimum of 1.3° and a maximum of 13.6° (*Figure 1—figure supplement 1*). The gazed person was either present in the video (original) or was erased from the video and appeared invisible. Therefore, in total, there were 80 videos x 2 (present vs. absent)=160 video stimuli. Of the person-present videos, half contained the target (40 videos) and half a distractor (40 videos) at the gazed goal locations.

To erase the gazed individuals from the images, we replaced the RGB values of pixels contained by the individual outline (annotated by research assistants) with the RGB values of those pixels of the immediate background (*Figure 1a–b*). The gazed person's location relative to the gazer's head had a mean of 6° visual angle, std = 3° visual angle (*Figure 1c*).

Across all the movies, the head regions subtended a mean size of 1.47° (std = 0.32°). Given that the average vertical length of the eyes spans 2.4 cm (0.024 m) (*Bekerman et al., 2014*) and the average vertical distance of the head is about 0.24 m (*Lee et al., 2006*), the eye only spanned.147° vertically (std = 0.032°).

## Procedure

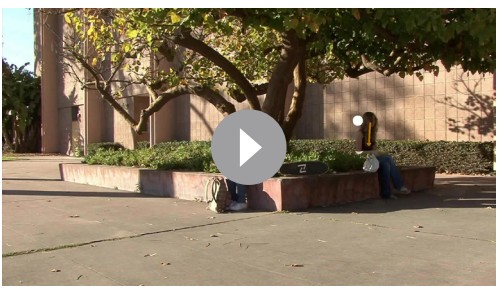

**Video 2.** Example trial with reverse saccade when the gazer was removed (white dot represents gaze position, yellow arrow represents gaze vector).
https://elifesciences.org/articles/83187/figures#video2

Subjects were asked to follow the gaze direction of the gazer as precisely as they can. Subjects were asked to respond if the target person was present or absent. There was a single target person for the entire experiment. Each participant finished sixteen practice trials to make sure they correctly followed the instructions to follow the gaze. During the practice, participants had unlimited time to familiarize themselves with pictures of the target person. The videos in the practice session were different from the actual experiment videos. Participants then completed the main experimental sessions after practice trials. During a session, observers completed all videos in

random order. In total, each observer finished 2 sessions x 160 trials/session = 320 trials. Participants first finished a nine-point calibration and validation. On each trial, the participants were instructed to fixate a cross and press the space bar to start the trial. If the eye tracker detected an eye movement away from the fixation cross of more than 1.5° visual angle when they pressed the space bar, the trial would not start, and participants were required to recalibrate and revalidate. The cross was located exactly at the location where the gazer's head would appear once the video started, so we can make sure the participants were looking straight at the gazer and observing the gazing behavior. During the video presentation (1.2 s), participants were asked to follow the gaze direction as precisely as they can. Once the video ended, the participants used a mouse to click if the target person was present or not (*Figure 1d*).

## AI model estimated gaze information

To quantify the gaze information in each video frame, we used a pre-trained deep neural network (DNN) based model (*Chong et al., 2020*), which makes an objective estimate of the gaze location for each video (*Figure 3a–b*). The model takes an entire image, a binary mask that defines a bounding box around the gazer's head location, and a cropped image of the gazer's head to produce a probability map of where the head's gaze is directed. We defined the model gaze estimation as the pixel location corresponding with the highest probability on the probability map. We repeated that for all the image frames from the video to obtain gaze estimation over time. To estimate the head angular velocity, we first took the difference in *angular displacement* for all continuous pairs of frames and smoothed the estimations by convolving the differences with a kernel size of 5 frames. Similarly, we calculated the head accelerations based on head velocity differences and smoothed them with a kernel size of 5.

## Human-estimated gaze information

Besides the AI model, we also recruited ten undergraduate research assistants to manually annotate the gazer vectors for all the video frames where the target or distractor was digitally erased. We used target/distractor-absent video frames for human annotations because we want to use isolated gaze goal direction information based on the gazer head direction without influences from peripheral information about potential gaze goals. We presented all the frames in random order. Annotators used Matlab to click on each image to draw the estimated gazer vector. We calculated the gaze estimation for each annotator and report the average estimation as the final human-estimated gazer vector for each frame. Human annotator estimates were consistent. Their gaze location estimation varied with mean standard error = 3.4° visual angle in the horizontal direction and 1.2° visual angle in the vertical direction (*Figure 3—figure supplement 1*).

## Forward and Reverse Saccades Detection

We defined a *forward saccade* as an eye movement in which the direction vector had a positive cosine similarity with the *gazer goal vector*. A *reverse saccade* was defined as a saccade vector that happened after a forward saccade and had a negative cosine similarity with the *gazer goal vector*. In addition, the reverse saccade endpoint was defined to be within a 2.5° visual angle from the gazer to differentiate them from corrective saccades that overshoot the gazer goal. A small subset of saccades was directed in the reverse direction because of the overshooting of the gaze-following saccade. The endpoints of these reverse saccades had a mean of 6.7° distance to the gazer's location. These saccades were considered different from reverse saccades to refixate the gazer and were not included in the analysis.

## Distribution of gaze information 200ms before the gazer's head stops

To compute the general head velocity range before the gazer stops the gaze behaviors, three annotators manually marked the time stamp when the gazer's head stops moving for each movie independently. We then defined the gazer stops timing as the average time across annotators for each movie. Finally, we calculated the mean gazer vector distance, angular displacement, head velocity, and head acceleration during the range of 200ms right before the gazer's head stops moving as the benchmarks (*Figure 3c*).

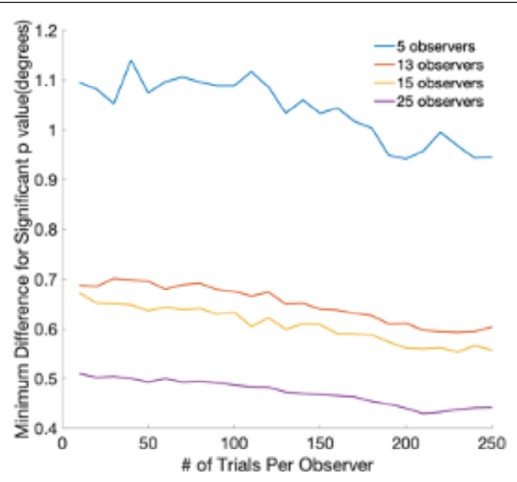

**Figure 7.** The minimum difference in degrees of visual angle that can be detected with a t-test at a significance level of 0.05. The graph shows the minimum detectable difference as a function of the number of trials per observer. Different lines correspond to different numbers of observers. Estimates are based on a database of 400 subjects and eye movements to faces.

## Statistical Analysis

We used within-subject ANOVA and t-tests for mean comparisons across different conditions. We also used bootstrap techniques to estimate the statistical significance of variations of saccade error (e.g., trials with reverse saccade vs. trials without reverse saccade) because of the non-normality of the distributions. To apply the bootstrap test, we sampled 25 participants with replacement and calculated the corresponding difference between conditions for each sampled subject (a bootstrap sample), and repeated the process 10,000 times. The distribution of resampled means or mean differences was used to assess statistical significance. All p values were corrected using a false discovery rate (FDR) to reduce the probability of making a Type I error. We used cluster-based permutation tests to compare the gazer's head velocity between trials with reverse saccades and those without reverse saccades. We computed the mean difference for each participant individually and permutated for 10,000 times. Based on corrected p values, we acquired time intervals with significant differences. We used Python to analyze all the data. For ANOVA tests we used package "pingouin"(*Vallat, 2018*). For the cluster-based permutation test, we used the package "MNE"(*Gramfort et al., 2013*).

## Statistical Power

To choose the sample size of our experiment, we used data from the previous work on eye movement to faces and estimated how many subjects would be needed to detect saccade endpoints that are about 1° visual angle apart. The database contained over 400 subjects making eye movements to faces (*Han et al., 2021a*). To find significant differences of 0.4–0.5° visual angle, we would need 25 subjects with 200 trials for experiments 1 and 2 (*Figure 7*). The goals of follow-up experiment 1 A and experiment 3 were to check whether reverse saccades occurred under two different conditions: (a) freezing the gazer, (b) changing the task so as not to instruct subjects to follow the gazer. We repeatedly resampled 5 participants and trials with replacements for 10,000 times from experiment 1. For each sample, we calculated the mean reverse saccades proportion to get 10,000 estimations. We estimated that if the proportion of reverse saccades remained the same as in Experiment 1, using 5 subjects would result in 98% of the time in a proportion of reverse saccades within the interval of 11–33% (mean ± standard deviation = 22 ± 11 %).

## Support Vector Machine (SVM) Models

For training SVM models, we first computed the proportion of trials in which observers executed reverse saccades for each movie. The median proportion of trials that included reverse saccades for all movies was 20%. We then did a median split of the movies into two groups. Those with a high proportion of reverse saccade (>20%) vs. those movies with a low proportion of the reverse saccade (<20%). Then we trained the SVM models with radial basis function kernel to classify whether a movie had a higher(>20 %) or lower (<20 %) probability of triggering reverse saccade. We trained leave-one-out SVM based on four gazer vector features: 1. Gazer vector distance 2. Gaze angular displacement 3. Head angular velocity 4. Head angular acceleration. For training each SVM model, we chose one of the four gaze features during a specific time range from the video onset as the predictor. We used the package "sklearn" for the training process (*Pedregosa et al., 2011*).

## Experiment 1A (random freeze)

### Subjects

Five undergraduate students (ages 18–20, 2 male, 3 female) from the University of California Santa Barbara were recruited as subjects for credits in this experiment. All have normal to corrected-to-normal vision. All participants signed consent forms to participate in the study. This experiment was conducted with fewer subjects because we were solely interested in quantifying whether reverse saccades occurred when freezing the movies. Using bootstrap resampling we estimated that if the proportion of reverse saccades remained the same as in Experiment 1, using 5 subjects would result in 98% of the time in a proportion of reverse saccades within a two standard deviation interval of 11–33% (22 ± 11 %).

### Experimental Setup and Stimuli

We had the same experiment stimuli and setup as experiment 1, except that we detected saccades during the movie presentation in real-time. When we detected the first gaze-following saccade, there was a 50% chance the gazer would be frozen (without movement) for the rest of the presentation time to prevent any gazer head motion that could potentially trigger a reverse saccade.

### Procedure

The procedure was the same as experiment 1. Participants were told to follow the gaze as precisely as they could during the movie presentation and to decide whether the target person was present in the video (50% present). In total, each observer finished 2 sessions x 160 trials/session = 320 trials. And participants were not aware of the random freezing of the video.

## Experiment 2

### Subjects

Twenty-five undergraduate students (ages 18–20, 10 male, 15 female) from the University of California Santa Barbara were recruited as subjects for credits in this experiment. All had normal to corrected-to-normal vision. All participants signed consent forms to participate in the study.

### Experimental Setup and Stimuli

We had the same experiment stimuli and setup as experiment 1, except that we detected reverse saccades during the movie presentation in real-time. When we detected a reverse saccade back to the gazer after the first gaze-following saccade, we digitally erased the gazer for the rest of the video in 50% of those trials. The digital deletion of the gazer before the reverse saccade's re-fixation prevented any foveal processing of the gazer.

### Procedure

The procedure was the same behavioral task as experiment 1. Participants were told to follow the gaze as precisely as they could during the movie presentation. Participants were unaware of the random digital erasure of the gazer.

## Experiment 3 (free-viewing search)

### Subjects

Five undergraduate students (ages 18–21, 3 male, 2 female) from the University of California Santa Barbara were recruited as subjects for credits in this experiment. All have normal to corrected-to-normal vision. All participants signed consent forms to participate in the study. This experiment was also conducted with fewer subjects (n=5) because we were solely interested in quantifying whether reverse saccades occurred during free viewing. As in experiment 1 a, we estimated that if the proportion of reverse saccades remained the same as in Experiment 1, using 5 subjects would result in 98% of the time in a proportion of reverse saccades within a two standard deviation interval of 11–33% (mean ± standard deviation = 22 ± 11 %).

## Experimental setup and stimuli

We used the same experiment stimuli and setup as in experiment 1.

## Procedure

The procedure was the same as experiment 1, except that we did not instruct participants to follow where the gazer looked at. Instead, we asked them to just free-viewing the video and respond whether the target person was present or absent. In total, each observer finished 2 sessions x 160 trials/session = 320 trials.

## Code availability

Code to replicate analysis is available at osf: https://osf.io/g9bzt/.

## Acknowledgements

The research was sponsored by the U.S. Army Research Office and was accomplished under Contract Number W911NF-19-D-0001 for the Institute for Collaborative Biotechnologies. MPE was supported by a Guggenheim Foundation Fellowship. The views and conclusions contained in this document are those of the authors and should not be interpreted as representing the official policies, either expressed or implied, of the U.S. Government. The U.S. Government is authorized to reproduce and distribute reprints for Government purposes notwithstanding any copyright notation herein.

## Additional information

### Funding

| Funder | Grant reference number | Author |
| --- | --- | --- |
| Army Research Office | W911NF-19-D-0001 | Miguel Patricio Eckstein |

The funders had no role in study design, data collection and interpretation, or the decision to submit the work for publication.

### Author contributions

Nicole Xiao Han, Conceptualization, Data curation, Software, Formal analysis, Validation, Investigation, Visualization, Methodology, Writing – original draft, Project administration, Writing – review and editing; Miguel Patricio Eckstein, Conceptualization, Resources, Supervision, Funding acquisition, Validation, Methodology, Writing – original draft, Project administration, Writing – review and editing

### Author ORCIDs

Nicole Xiao Han ⓘ http://orcid.org/0000-0003-2860-2743

### Ethics

The experiment protocol was approved by the University of California Internal Review Board with protocol number 12-22-0667. All participants signed consent forms to participate in the experiment and to include their images in resulting publications.

### Decision letter and Author response

Decision letter https://doi.org/10.7554/eLife.83187.sa1
Author response https://doi.org/10.7554/eLife.83187.sa2

## Additional files

### Supplementary files

• MDAR checklist

### Data availability

All data generated or analyzed during this study are deposited at https://osf.io/g9bzt/.

The following dataset was generated:

| Author(s) | Year | Dataset title | Dataset URL | Database and Identifier |
|---|---|---|---|---|
| Han NX | 2022 | Eye Movement Planning During Dynamic Gaze Following | https://osf.io/g9bzt/ | Open Science Framework, g9bzt |

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
