## [Editor Report]

This important work substantially advances our understanding of how human eye movements are shaped by social cues. Using clever experimental manipulations and innovative artificial intelligence analysis tools, the paper identifies distinctive patterns of saccadic eye movements tracking another person's gaze during dynamic video-scene viewing. This work will be of broad interest to psychologists, biologists, and neuroscientists interested in human social behavior.

---

## [Decision Letter]

**Decision letter after peer review:**

Thank you for submitting your article "Inferential Eye Movement Control while Following Dynamic Gaze" for consideration by *eLife*. Your article has been reviewed by 3 peer reviewers, and the evaluation has been overseen by a Miriam Spering as Reviewing Editor and Joshua Gold as the Senior Editor. The reviewers have opted to remain anonymous.

Essential revisions:

1) Please clarify whether cognitive expectations about the role of the second person in videos might have biased gaze patterns and explain the difference between the present/absent conditions (see Reviewer 1).

2) Discuss the choice of creating "absent videos" by digitally removing the target/distractor person from the video, and how this might have affected saccade latency and amplitude variability (Reviewer 1).

3) Consider additional analyses suggested by Reviewer 1.

4) Please explicitly state what observers were asked to do (Methods and repeat in Results); see Reviewer 3.

5) Add requested literature to the introduction and clarify some terminology throughout, please see below.

*Reviewer #1 (Recommendations for the authors):*

Discussion: The limitations/features of the stimulus material and their potential impact on the results should be considered in the discussion.

The relative contribution of foveal and peripheral information: The authors did a great job in analyzing the relative contribution of foveal and peripheral information, but there might be even further potential for separation: Foveal information on the head direction of the gazer should be quite informative about the direction of the gaze goal, but less about its distance (at least in a 2D plane) and participants might need to rely on peripheral information on potential gaze goals to determine the correct distance. This might influence the error in the saccade direction and saccade amplitude differently. Information about amplitude might be especially uncertain in absent trials where no or multiple objects are in the line of sight of the gazer. I think it might be informative to separately analyze errors in saccade amplitude and direction because they should be related to different pieces of information.

Further analysis: I was wondering to what extent the necessary precision of information about the gazer's head direction depends (a) on the distance between the gazer and the gaze goal and (b) on the nature of the gaze goal. One could expect that higher precision is necessary when the distance between the gazer and the goal is large. For instance, the authors could test in their data how the distance between the gazer and gaze goal influences the occurrence of reverse saccades and the error of the pre- and post-reverse saccades. I admit that this is a bit tangential, but it would be informative as to whether the eye movement pattern is sensitive to the required information for the task. Higher precision might also be necessary when the gaze goal is small or when it is composed of meaningful patterns like a human. For instance, a 10 cm difference in gaze location doesn't matter much when someone is looking at a tree, but it is highly relevant to determine if someone is looking another person in the eyes or not. This aspect is clearly out of the scope of the present study but might be interesting for future studies.

Terminology: I stumbled over the term "reverse" saccades. This seems unnecessarily unspecific because the saccades not only reverse the direction but actually return to the previous fixation location on the gazer. Hence, "return" saccades might be more precise.

Line 77: The difference between the central and peripheral visual fields not only concerns the retina and the cortex, but also subcortical areas like the LGN or the SC.

Line 100: How can gaze cueing be a correlate of ASD? Impairments or alterations in gaze cueing can be correlates of ASD.

Line 158: It is not clear how the error of the first fixation is calculated in absent trials. If there is no person at the gaze goal (or no gaze goal at all), it doesn't seem to be meaningful to calculate an error (at least in terms of distance, the direction might still be well defined by the head direction of the gazer).

Line 203: As the authors describe later in the discussion, the programming of upcoming saccades cannot be altered during the saccadic dead time. The saccadic dead time does not seem to be considered when calculating the gazer vector angular error at the time of the saccade. When one would take the angular error some 50 ms before saccade initiation into account for the dead time, the advantage of prediction would be even larger.

Line 466: Does the fact that the occurrence of reverse saccades depends on the gazer's head velocity imply that humans have some expectation about typical head rotation speeds and assume that a below-average speed indicates the end of the head rotation?

Lines 628 and 644: How well did the annotators agree?

*Reviewer #3 (Recommendations for the authors):*

I have overall really appreciated this work and I have little to say. My main comments are likely related to some theoretical aspects that could be improved. Indeed, my feelings during my own reading were that this work focuses more on technical aspects and analyses (which is a good point) but less on what is the general meaning that this work provides.

The introduction could be improved a bit. For instance, in lines 98-102, I would also make explicit mentions of some recent reviews on gaze following and gaze cueing (Capozzi and Ristic, 2018; Dalmaso et al., 2020; McKay et al., 2021), to provide naïve readers with a more complete picture about these phenomena.

Then, from line 104: it is true that most studies used static images or faces in isolation, but there are much more exceptions. I am thinking about studies with real social interactions (Lachat et al., 2012; Macdonald and Tatler, 2013) or in multi-agent contexts (e.g., Sun et al., 2017), which could be reported for completeness.

As for theory, if I am right the task required to produce a saccade and then to identify whether a person was the correct target or just a distractor; in some cases, no person at all was presented (except the gazer). What I am missing here is a further condition in which the gazer looks towards a non-social target, such as an object. I am wondering if a different pattern of results could be expected when a target is an object and not another human, as we know that faces and people, when used as targets, can shape eye movements peculiarly s compared to non-social stimuli (Mares et al., 2016). Perhaps I'm missing something obvious here.

References

Capozzi, F., and Ristic, J. (2018). How attention gates social interactions. Annals of the New York Academy of Sciences, 1426(1), 179-198. https://doi.org/10.1111/nyas.13854

Dalmaso, M., Castelli, L., and Galfano, G. (2020). Social modulators of gaze-mediated orienting of attention: A review. Psychonomic Bulletin and Review, 27(5), 833-855. https://doi.org/10.3758/s13423-020-01730-x

Lachat, F., Conty, L., Hugueville, L., and George, N. (2012). Gaze Cueing Effect in a Face-to-Face Situation. Journal of Nonverbal Behavior. https://doi.org/10.1007/s10919-012-0133-x

Macdonald, R. G. R., and Tatler, B. B. W. (2013). Do as eye say: Gaze cueing and language in a real-world social interaction. Journal of Vision, 13(4), 1-12. https://doi.org/10.1167/13.4.6

Mares, I., Smith, M. L., Johnson, M. H., and Senju, A. (2016). Direct gaze facilitates rapid orienting to faces: Evidence from express saccades and saccadic potentials. Biological Psychology, 121, 84-90. https://doi.org/10.1016/j.biopsycho.2016.10.003

McKay, K. T., Grainger, S. A., Coundouris, S. P., Skorich, D. P., Phillips, L. H., and

Henry, J. D. (2021). Visual attentional orienting by eye gaze: A meta-analytic review of the gaze-cueing effect. Psychological Bulletin, 147(12), 1269-1289. https://doi.org/10.1037/bul0000353

Sun, Z., Yu, W., Zhou, J., and Shen, M. (2017). Perceiving crowd attention: Gaze following in human crowds with conflicting cues. Attention, Perception, and Psychophysics, 1-11. https://doi.org/10.3758/s13414-017-1303-z

[Editors’ note: further revisions were suggested prior to acceptance, as described below.]

Thank you for resubmitting your work entitled "Inferential Eye Movement Control while Following Dynamic Gaze" for further consideration by *eLife*. Your revised article has been evaluated by Joshua Gold (Senior Editor) and Miriam Spering (Reviewing Editor).

The manuscript has been improved and is almost ready to be accepted, pending just one remaining issue regarding your angular error analysis that needs to be addressed, as outlined in the reviewer's comments below:

*Reviewer #1 (Recommendations for the authors):*

The authors considerably improved the manuscript and I just have a few remaining comments:

Analysis of saccade errors: I commend the authors for including the angular error of saccades, but I am bit puzzled why they kept the Euclidian distance as the second error metric. The Euclidian distance includes the angular error and therefore the two measures are not completely orthogonal to each other. I would have analyzed the amplitude error in addition to the angular error to have two completely orthogonal measures. Looking at the heat maps in Figure 2, it seems that the deletion of the target/distractor predominantly leads to more elongated landing distributions along the horizontal (gaze) direction. The current way of reporting saccade errors doesn't seem to reflect this increase in anisotropy.

Terminology: I understand that the authors prefer "reverse" over "return" saccades because it matches better their exploration of the data and the narrative in the manuscript. My perspective, however, was about how other researchers will refer to that effect in future studies and in that sense a more precise terminology might be better for future referencing. Ultimately, it's the authors' decision which perspective is more important to them.

Saccadic dead time: estimations of saccadic dead time can vary quite a bit and therefore my suggestion was to use a very conservative estimate at the lower end of the range to avoid exaggerating the effect. However, I'm fine with the author's choice of a more average value of 100 ms.

---

## [Author Response]

Essential revisions:1) Please clarify whether cognitive expectations about the role of the second person in videos might have biased gaze patterns and explain the difference between the present/absent conditions (see Reviewer 1).

We have now added a paragraph (page 24-25) addressing how cognitive expectations might influence the current results. We also discuss other limitations such as how our measurements might vary with the visibility of the gaze goal. In addition, one reviewer also brought up the interesting question of whether the observers’ gaze following behavior might vary for social targets (other people) vs. for non-social inanimate objects. We have also mentioned this as an interesting future possible investigation on page 25.

2) Discuss the choice of creating "absent videos" by digitally removing the target/distractor person from the video, and how this might have affected saccade latency and amplitude variability (Reviewer 1).

We now spell out our motivations for this manipulation, how it relates to more ecologically valid situations and its effects on saccades (page 4). There is an entire paragraph explaining the rationale. We also describe the rationale below in our detailed response to the reviewers.

3) Consider additional analyses suggested by Reviewer 1.

We now incorporated the suggested analyses by Reviewer 1 including the angular error (see new Figures 2d, Figure 5e, and text references throughout the Results section).

4) Please explicitly state what observers were asked to do (Methods and repeat in Results); see Reviewer 3.

We have expanded the description of what the observers were asked to do in the Methods (page 26), the Results section (see page 5). We have also edited the timeline in Figure 1a and the accompanying caption.

5) Add requested literature to the introduction and clarify some terminology throughout, please see below.

We have incorporated the suggested literature in the introduction and discussion.

Reviewer #1 (Recommendations for the authors):Discussion: The limitations/features of the stimulus material and their potential impact on the results should be considered in the discussion.The relative contribution of foveal and peripheral information: The authors did a great job in analyzing the relative contribution of foveal and peripheral information, but there might be even further potential for separation: Foveal information on the head direction of the gazer should be quite informative about the direction of the gaze goal, but less about its distance (at least in a 2D plane) and participants might need to rely on peripheral information on potential gaze goals to determine the correct distance. This might influence the error in the saccade direction and saccade amplitude differently. Information about amplitude might be especially uncertain in absent trials where no or multiple objects are in the line of sight of the gazer. I think it might be informative to separately analyze errors in saccade amplitude and direction because they should be related to different pieces of information.

This is a very interesting and insightful point. It would make sense that the peripheral information informed the saccade distance more than the saccade angular error. To investigate the issue, we have now added saccade angular error to all our analyses. These analyses include angular error of first saccades with and without the gaze goal (digital deleting of the gaze goal; Figure 2d); angular error of forward saccade pre and post-reverse saccade (Figure 6b), angular error of forward saccade with gazer removed vs. not removed (Figure 6c).

Here is the interesting and even perhaps surprising result. Our findings show consistency across both measures. The peripheral information about the gaze goal benefited in reducing both the saccade Euclidean error and the saccade angular error. The explanation might lie in the anticipatory nature of the observers’ saccades. The gazer’s head direction is typically pointing about 22 to 28 degrees off from the gaze goal when the observer makes the saccade. Thus, the observer is still needing to make an inference of where the head will point to at the end of the head movement when programming the saccade. Eliminating the peripheral visual information about a potential gaze goal results in an increased error about the gazer’s final head direction.

The rationale (as pointed out by the reviewer!) for a different result for the saccade endpoint Euclidean error vs. the saccade angular error is now explained on page 5. The possible explanation for the results is now discussed on page 21.

Thanks for suggesting the analysis.

Further analysis: I was wondering to what extent the necessary precision of information about the gazer's head direction depends (a) on the distance between the gazer and the gaze goal and (b) on the nature of the gaze goal. One could expect that higher precision is necessary when the distance between the gazer and the goal is large. For instance, the authors could test in their data how the distance between the gazer and gaze goal influences the occurrence of reverse saccades and the error of the pre- and post-reverse saccades. I admit that this is a bit tangential, but it would be informative as to whether the eye movement pattern is sensitive to the required information for the task. Higher precision might also be necessary when the gaze goal is small or when it is composed of meaningful patterns like a human. For instance, a 10 cm difference in gaze location doesn't matter much when someone is looking at a tree, but it is highly relevant to determine if someone is looking another person in the eyes or not. This aspect is clearly out of the scope of the present study but might be interesting for future studies.

We appreciate the thoughtful comments about the role of the eccentricity of the gaze goal, its size and visibility in the periphery. We agree that these are really interesting questions. How the studied effects (saccade error, reverse saccades) vary with systematic changes in gaze goal object size and task requirements (fixating a larger tree vs. a person’s face) is an interesting question. We believe that the real-world video dataset might not be ideal for answering such questions because they do not manipulate each of these variables while controlling for the other.

However, we did analyze the role of eccentricity of the gaze goal in detail which we deemed could be studied to some extent in our data. Here are some of the results below.

– Distance between gazer and gaze goal (eccentricity of gaze goal): We computed the distance between the gazer goal to the gazer measured in terms of degrees and relative to the fovea of the observer (fixating on the gazer). Author response image 1 is the histogram of these eccentricities. For the subsequent analyses, we did a median split of gaze goal eccentricities: median = 4.93° visual angle; low eccentricity (<4.93°) vs. high eccentricity (=4.93°).

**Author response image 1. sa2fig1:** Histogram of eccentricity (distance between) gazer head and the gaze goal.

– Saccade properties for low vs. high eccentricity gaze goals: As expected the observer saccade amplitudes to high eccentricity gaze goals were larger than for low eccentricity gaze goals (=4.93° for high eccentricity vs. <4.93° for low eccentricity). There was no difference in the 1st saccade latency with eccentricity.

– Saccade distance error and angular error vs. gaze goal eccentricity: We computed the error vs. gaze goal eccentricity for experiment 1 and found that there was a similar angular error for the forward saccade of gaze goals at low and high eccentricities (low 14.5 deg vs. high 11.5 deg, p = 0.28 ), but a significantly higher saccade Euclidean error for gaze goals with high eccentricity (low 1.23° vs. high 2.21°, p < 1e-05). Author response image 3 shows the effect of eccentricity.

**Author response image 3. sa2fig3:** 

– Interaction of eccentricity with the influence of reverse saccades on subsequent fixation error (pre- vs. post-reverse saccade errors): We analyzed how gaze goal eccentricity interacted with the benefit of reverse saccades on subsequent forward saccade errors. We found that both the saccade Euclidean error and the angular error were reduced after the reverse saccade for *high eccentricity* gaze goals. This seems like an intuitive result. High eccentricity gaze goals have more ambiguity as there are more intervening potential objects between the gazer and the actual gaze goal.Although the various findings are interesting, right now, we did not include these analyses in the revised version of the paper. We felt that investigating these issues required more controlled stimuli that isolated the influence of eccentricity. As an example, the high and low eccentricity gaze goal videos contain different gazer head movements with different head amplitude movements and head velocities. It is difficult to confidently conclude that the differences in observers’ eye movements behavior are solely related to the gaze goal eccentricity. The ideal stimuli to test the influence of gaze-goal eccentricity would be to digitally insert targets along the gaze vectors for the same videos. That would keep the gazer head information the same but vary the gaze goal eccentricity of the videos Right now this is not the case. In addition, we felt that the paper has so many results that adding this would be distracting.

Of course, if the reviewer feels different and wants to see these analyses in the paper, we will be happy to include the results in the supplementary material!

Terminology: I stumbled over the term "reverse" saccades. This seems unnecessarily unspecific because the saccades not only reverse the direction but actually return to the previous fixation location on the gazer. Hence, "return" saccades might be more precise.

Indeed, the reverse saccades do return to the gazer’s head and the term “return” saccades might be more precise. Here is why we use “reverse” saccades.

The paper's narrative follows that of our scientific investigation with the data set. Our analyses found these surprising saccades in the opposite direction but we did not know where these were directed to. Further analyses revealed that observers were re-fixating the gazer’s head. The paper follows that narrative. It identifies the reverse saccades, analyzes their timing and then in a subsequent section investigates where they are directed to. If we called them “return” saccades from the set it would anticipate the result we are uncovering in the section titled: “Functional role of reverse saccades”

That is our rationale to call them “reverse” saccades in the paper. Our preference is to keep it with this narrative. What we have done is insert a sentence in the Discussion section (page 22 line 543) that the reverse saccades could be more precisely described as return saccades. This is inserted in the section (“Functional role of reverse saccades”) after showing that the saccades are re-fixating the gazer’s head. If the reviewer feels a strong preference for changing everything to return saccades we will do so.

Line 77: The difference between the central and peripheral visual fields not only concerns the retina and the cortex, but also subcortical areas like the LGN or the SC.

We have added this information and references to LGN and SC over-representation of the foveal region. Thank you.

Line 100: How can gaze cueing be a correlate of ASD? Impairments or alterations in gaze cueing can be correlates of ASD.

It cannot. Thanks for the correction. We have inserted the word “impairments” as suggested by the reviewer.

Line 158: It is not clear how the error of the first fixation is calculated in absent trials. If there is no person at the gaze goal (or no gaze goal at all), it doesn't seem to be meaningful to calculate an error (at least in terms of distance, the direction might still be well defined by the head direction of the gazer).

It might seem counterintuitive but there is a “gold standard” or “truth” about where the gazer is looking at. It is correct that there is some ambiguity in the images. Even when the gaze goal is present, the person might be looking at a small object that is difficult for the observer to detect peripherally.

We eliminated the gaze goal, so as the reviewer correctly points out, there will be increased ambiguity about where the gazer is looking at. The observer will not have the peripheral visual signal to guide the eye movement and will have to more heavily on the head direction of the gazer.

Line 203: As the authors describe later in the discussion, the programming of upcoming saccades cannot be altered during the saccadic dead time. The saccadic dead time does not seem to be considered when calculating the gazer vector angular error at the time of the saccade. When one would take the angular error some 50 ms before saccade initiation into account for the dead time, the advantage of prediction would be even larger.

Thoughtful comment. We did consider this. But, in the end, we preferred to be conservative and did not add the dead time which would amplify the advantage of saccade prediction. Arguably, having done so would favor our hypothesis. Thus, we felt that including the dead time into our estimation of anticipatory would be perceived by reviewers/readers as convenient to amplify the claim of “anticipatory/inferential eye movements”. Our estimate of the dead time from our studies based on reverse correlation (Caspi et al., 2004) is a little longer than what the reviewer mentions 100 ms. We now mention the calculation with a 100 ms dead time in the Discussion section. If the reviewer feels the evidence for 50 ms is stronger (please provide a reference if this is the case) we will change our numbers.

We are hesitant to include the estimates with the dead time in the abstract and do not have room to explain the nuisance of the two estimates. But if the reviewer feels strongly about this, we will.

Line 466: Does the fact that the occurrence of reverse saccades depends on the gazer's head velocity imply that humans have some expectation about typical head rotation speeds and assume that a below-average speed indicates the end of the head rotation?

Yes, this is what we are suggesting. The analysis of typical gazer head rotations speeds 200 ms before the head stops are similar to the average speed before the reverse saccades (Figure 3c). We now emphasize the reviewer’s point in the discussion so it comes across more clearly (page 22).

Lines 628 and 644: How well did the annotators agree?

We have incorporated into the supplementary material some basic analyses about the annotators’ agreement. The average (across videos) of the standard deviations of the estimation of the gaze goal locations across the ten annotators were: 3.4° horizontally and 1.2° vertically. The average standard deviation across estimated gaze goal angles (between the gazer’s head and estimated gaze goal) of the annotators was: 11.6 deg.

Supplementary Figure S5. includes the full distribution (across videos) of standard deviations (across annotators) of location estimations of the gaze goals. We also included the full distribution of standard deviations of estimated angles. See below for reference.

Reviewer #3 (Recommendations for the authors):I have overall really appreciated this work and I have little to say. My main comments are likely related to some theoretical aspects that could be improved. Indeed, my feelings during my own reading were that this work focuses more on technical aspects and analyses (which is a good point) but less on what is the general meaning that this work provides.The introduction could be improved a bit. For instance, in lines 98-102, I would also make explicit mentions of some recent reviews on gaze following and gaze cueing (Capozzi and Ristic, 2018; Dalmaso et al., 2020; McKay et al., 2021), to provide naïve readers with a more complete picture about these phenomena.

We appreciate the reviewer mentioning these. We admit to being new to this subfield and we try to be comprehensive with our literature review. But, we clearly have missed relevant literature. We have now incorporated these references. Thanks for the pointers.

Then, from line 104: it is true that most studies used static images or faces in isolation, but there are much more exceptions. I am thinking about studies with real social interactions (Lachat et al., 2012; Macdonald and Tatler, 2013) or in multi-agent contexts (e.g., Sun et al., 2017), which could be reported for completeness.

Thanks. Appreciated. We have incorporated the reviewer’s suggestions.

As for theory, if I am right the task required to produce a saccade and then to identify whether a person was the correct target or just a distractor; in some cases, no person at all was presented (except the gazer). What I am missing here is a further condition in which the gazer looks towards a non-social target, such as an object. I am wondering if a different pattern of results could be expected when a target is an object and not another human, as we know that faces and people, when used as targets, can shape eye movements peculiarly s compared to non-social stimuli (Mares et al., 2016). Perhaps I'm missing something obvious here.

This is another good point. The question of how the dynamics of the gaze-following eye movements vary when there is a person vs. a non-social target is very interesting. We do not have the data to answer this question but it is certainly and interesting question. Such comparison would have to be implemented by control of the visibility of the social and non-social gaze goals to eliminate low-level confounds. We have included a statement in the discussion (page 24) related to these interesting remaining questions.

[Editors’ note: what follows is the authors’ response to the second round of review.]

The manuscript has been improved and is almost ready to be accepted, pending just one remaining issue regarding your angular error analysis that needs to be addressed, as outlined in the reviewer's comments below:

Reviewer #1 (Recommendations for the authors):The authors considerably improved the manuscript and I just have a few remaining comments:Analysis of saccade errors: I commend the authors for including the angular error of saccades, but I am bit puzzled why they kept the Euclidian distance as the second error metric. The Euclidian distance includes the angular error and therefore the two measures are not completely orthogonal to each other. I would have analyzed the amplitude error in addition to the angular error to have two completely orthogonal measures. Looking at the heat maps in Figure 2, it seems that the deletion of the target/distractor predominantly leads to more elongated landing distributions along the horizontal (gaze) direction. The current way of reporting saccade errors doesn't seem to reflect this increase in anisotropy.

Thanks for the thoughtful comment and suggestion. We have added the new figures and analyses on saccade amplitude error to the manuscript (new figure 2,5,6) so that we have a comprehensive understanding of different dimensions of the error measurements (angular error and amplitude error). In addition, we chose to keep the Euclidean error (which combines both angular and amplitude error) because it is easy to interpret. In Figure 2 we show a where we now have all three errors.

Based on the analyses, the saccade amplitude error (absolute difference between saccade amplitude and the distance between the gazer and gaze goal location) is very similar to Euclidean error (distance between the saccade endpoint to the gaze goal location).

Terminology: I understand that the authors prefer "reverse" over "return" saccades because it matches better their exploration of the data and the narrative in the manuscript. My perspective, however, was about how other researchers will refer to that effect in future studies and in that sense a more precise terminology might be better for future referencing. Ultimately, it's the authors' decision which perspective is more important to them.

After some thought, for the sake of the narrative of the paper, we would like to keep the word “reverse saccade” up to the point where we identified that these “reverse saccades” were aimed at the gazer’s heads. We bring up the concept of referring to “reverse saccades” as “return saccades” in the discussion and it might be something we might adopt moving forward.

Saccadic dead time: estimations of saccadic dead time can vary quite a bit and therefore my suggestion was to use a very conservative estimate at the lower end of the range to avoid exaggerating the effect. However, I'm fine with the author's choice of a more average value of 100 ms.

Thanks for the feedback!